# Pathophysiological Basis for Nutraceutical Supplementation in Heart Failure: A Comprehensive Review

**DOI:** 10.3390/nu13010257

**Published:** 2021-01-17

**Authors:** Vincenzo Mollace, Giuseppe M. C. Rosano, Stefan D. Anker, Andrew J. S. Coats, Petar Seferovic, Rocco Mollace, Annamaria Tavernese, Micaela Gliozzi, Vincenzo Musolino, Cristina Carresi, Jessica Maiuolo, Roberta Macrì, Francesca Bosco, Marcello Chiocchi, Francesco Romeo, Marco Metra, Maurizio Volterrani

**Affiliations:** 1Department of Health Sciences, Institute of Research for Food Safety & Health, University “Magna Graecia” of Catanzaro, 88100 Catanzaro, Italy; rocco.mollace@gmail.com (R.M.); an.tavernese@gmail.com (A.T.); micaela.gliozzi@gmail.com (M.G.); xabaras3@hotmail.com (V.M.); carresi@unicz.it (C.C.); jessicamaiuolo@virgilio.it (J.M.); robertamacri85@gmail.com (R.M.); francesca-bosco88@libero.it (F.B.); 2Cardiology Clinical Academic Group, St George’s Hospitals NHS Trust University of London, London SW17 0QT, UK; giuseppe.rosano@gmail.com; 3Department of Cardiology, IRCCS San Raffaele Pisana, 00166 Rome, Italy; andrewjscoats@gmail.com (A.J.S.C.); maurizio.volterrani@sanraffaele.it (M.V.); 4Department of Cardiology, Charité–Universitätsmedizin Berlin, 10117 Berlin, Germany; s.anker@cachexia.de; 5Faculty of Medicine, Belgrade University, 11000 Belgrade, Serbia; seferovic.petar@gmail.com; 6Department of Experimental and Applied Medicine, Institute of Cardiology, University of Brescia, 25121 Brescia, Italy; metramarco@libero.it; 7Department of Diagnostic Imaging and Interventional Radiology, Policlinico Tor Vergata, 00199 Rome, Italy; marcello.chiocchi@gmail.com; 8Department of Experimental Medicine, University of Rome “Tor Vergata”, 00199 Rome, Italy; romeocerabino@gmail.com

**Keywords:** heart failure, dysfunctional cardiomyocytes, patho-physiological mechanisms, oxidative stress, nutraceuticals

## Abstract

There is evidence demonstrating that heart failure (HF) occurs in 1–2% of the global population and is often accompanied by comorbidities which contribute to increasing the prevalence of the disease, the rate of hospitalization and the mortality. Although recent advances in both pharmacological and non-pharmacological approaches have led to a significant improvement in clinical outcomes in patients affected by HF, residual unmet needs remain, mostly related to the occurrence of poorly defined strategies in the early stages of myocardial dysfunction. Nutritional support in patients developing HF and nutraceutical supplementation have recently been shown to possibly contribute to protection of the failing myocardium, although their place in the treatment of HF requires further assessment, in order to find better therapeutic solutions. In this context, the Optimal Nutraceutical Supplementation in Heart Failure (ONUS-HF) working group aimed to assess the optimal nutraceutical approach to HF in the early phases of the disease, in order to counteract selected pathways that are imbalanced in the failing myocardium. In particular, we reviewed several of the most relevant pathophysiological and molecular changes occurring during the early stages of myocardial dysfunction. These include mitochondrial and sarcoplasmic reticulum stress, insufficient nitric oxide (NO) release, impaired cardiac stem cell mobilization and an imbalanced regulation of metalloproteinases. Moreover, we reviewed the potential of the nutraceutical supplementation of several natural products, such as coenzyme Q10 (CoQ10), a grape seed extract, *Olea Europea* L.-related antioxidants, a sodium–glucose cotransporter (SGLT2) inhibitor-rich apple extract and a bergamot polyphenolic fraction, in addition to their support in cardiomyocyte protection, in HF. Such an approach should contribute to optimising the use of nutraceuticals in HF, and the effect needs to be confirmed by means of more targeted clinical trials exploring the efficacy and safety of these compounds.

## 1. Introduction

Heart failure (HF) represents a multifactorial disease state with a global prevalence of 1–2% in the population [1]. Comorbidities frequently associated with HF are hypertension, diabetes, and obesity or hyperlipidaemia. These comorbidities thereby increase the prevalence of the disease, the rate of hospitalisation, and mortality [2,3]. In addition, ageing affects the number of patients experiencing HF; the occurrence of the disease rises to 10% in people aged >70 years [4,5].

According to recent international treatment guidelines, better management of risk factors and increased pharmacological and non-pharmacological treatment significantly reduce the impact of HF and its consequences. In particular, recent data from a pilot study on HF performed by the European Society of Cardiology (ESC) showed that the all-cause mortality prevalence, at 12 months, was 17% for hospitalised HF patients and 7% for stable/ambulatory HF patients, with hospitalisation rates for the same period being 44% and 32%, respectively [1]. Hence, several relevant outcomes of HF remain unsatisfactory and additional studies are required to address the currently unmet needs in the management of the disease. 

Recently, growing evidence has suggested that greater control of the nutritional balance in patients experiencing HF (with special regards to the micronutrient and nutraceutical supply) leads to a significant improvement in the symptoms and consequences of the disease [5,6,7]. On the other hand, the identification of novel biomolecular mechanisms that lead to the onset of HF, which are only partially counteracted by current treatments, suggests the potential for an additional contribution in the management of HF via optimal micronutrient and nutraceutical supplementation [8,9,10,11,12]. 

There is evidence that the majority of HF patients exhibit an insufficient support of micronutrients and that such a condition leads to an increased prevalence of the disease [13,14,15,16]. Moreover, conditions of dysfunctional myocardium and impaired muscle functionality are accompanied by oxidative stress and micronutrient deficiency [13,14,15,16,17]. Finally, there is also evidence that an insufficient micronutrient supply may trigger HF, as well as impair a pre-existing condition of cardiac dysfunction [13,17].

Additionally, multiple conditions related to micronutrient deficiency can lead to a depletion of the lean body mass (LBM). A pro-inflammatory state, inadequate intake, metabolic changes, increased oxidative stress and increased nutrient loss can all contribute to a loss of LBM. Furthermore, the depletion of LBM can affect vital organs, including the myocardium, thereby negatively influencing its functional capacity [12,13,14,15,16].

Although preliminary evidence suggests that nutraceutical supplementation could be beneficial in treating patients undergoing a decaying heart performance, there is limited evidence supporting its extensive use in patients with HF, as reported in recent studies and meta-analyses [18,19] (Table 1). The major weaknesses in the existing evidence include the small number of studies performed and the inconsistent number of study participants, which is associated with a diminished study quality with a high risk of bias. Considering this, definite conclusions cannot be drawn. Consequently, it is the general opinion that clinicians should presently favour other treatments that have clearly been shown to decrease mortality [18,19,20] (Table 2).

Therefore, although the use of nutraceutical supplementation is of interest in the prevention and treatment of cardiovascular disease, the potential for its use in the treatment of HF still needs to be adequately assessed. More studies are required to verify the optimal use of nutraceutical supplementation in counteracting myocardial dysfunction in HF patients.

The aim of this review article is to assess the pathophysiological mechanisms involved in the early stages of HF and the potential for timely nutraceutical supplementation to support the failing myocardial cells in tandem with the pharmacological and non-pharmacological interventions currently being used. Moreover, a further objective is to optimise nutraceutical intervention through the identification of candidate natural extracts to be used in future clinical studies in patients experiencing HF.

## 2. Emerging Pathophysiological Mechanisms Involved in the Onset of HF

### 2.1. Energy Deficiency and Mitochondrial Impairment in HF

In patients with HF, energetic deficits are palpable and can be detected, both in vivo and in vitro, in the early stages of the disease [94]. In particular, it has been reported that non-invasive measurement of the myocardial phosphocreatine (PCr) via 31P-magnetic resonance spectroscopy (MRS) and an assessment of the PCr to adenosine triphosphate (ATP) ratio indicate an energy shortage, which is associated with many cardiac disease states, including ischemic heart disease and cardiomyopathies [95]. In this context, the degradation of ATP, which is frequently associated with diastolic dysfunction, represents a biomolecular target of HF with a preserved ejection fraction (HFpEF) and indicates the occurrence of an increased energetic demand and/or energetic mismatch in HF [96,97,98,99,100,101]. This occurs at the early stages of the disease, thereby showing that counteracting the energy deficit may be crucial in the development of HF, though the mechanisms of energy impairment need to be better assessed.

Several studies, ranging from those conducted 50 years ago at the National Institute of Health (NIH) to more recent studies [102,103,104,105], have suggested that the electron transport chain (ETC) function is not impaired in failing versus not-failing hearts, although conflicting data exist [106]. Studies on the role of Krebs cycle activity, which is responsible for the production of nicotinamide adenine dinucleotide (NADH) and flavin adenine dinucleotide (FADH2) from acetyl-coenzyme A, have shown an impairment of this energy-producing pathway in decaying hearts [104,107]. This effect is accompanied by an impaired spatial pattern, which characterises mitochondria within cardiac cells. In particular, energy depletion affects mitochondrial functionality. This is shown by the expression of a class of fusion proteins, such as mitofusins (Mfn1 and Mfn2), which also potentially affect the Sarcoplasmic Reticulum (SR)-mitochondrial Ca^2+^ microdomain [108,109,110], and this, in turn, is accompanied by dysregulation of the mitochondrial Ca^2+^ uniporter (MCU), seen by means of a mouse model of pressure overload [111].

Together, these data indicate that in HF, Ca^2+^-induced stimulation of the Krebs cycle is impaired by decreased mitochondrial Ca^2+^ uptake during cardiac workload transitions.

Mitochondrial functionality is also crucial in the balance between energy production and the consistent anti-oxidative capacity of failing cardiomyocytes [112,113]. In particular, it is well-known that the oxidation of NADH, which occurs during an elevated cardiac workload, is an important consequence of this energy supply and demand mismatch [114]. The depleted NADPH-coupled anti-oxidative capacity is then overwhelmed by reactive oxygen species (ROS) production by NADH-coupled respiration in the electron transport chain (ETC) [115,116]. Consequently, this favours the reverse mode of the mitochondrial nicotinamide nucleotide transhydrogenase (Nnt), thus dissipating the anti-oxidative capacity [115]. Aon et al. proposed that “mitochondria have been evolutionarily optimized to maximize energy output while keeping ROS overflow to a minimum by operating in an intermediate redox state” [116].

Therefore, extreme oxidation, as aforementioned, and a reduction in the mitochondrial redox state, such as that which occurs during ischaemia, are to be avoided in order to achieve optimal conditions for cardiac mitochondria. However, the working heart constantly produces adenosine diphosphate (ADP), physiologically accelerating respiration and increasing oxidation in the respiratory chain. Therefore, it is unlikely that increased oxidative stress in HF is solely due to a net increase in ROS production, but rather due to a diminished ROS-scavenging capacity [117]. This leads to a vicious circle, increasing oxidative stress by exacerbating the mismatch in the energy supply and demand [118]. Mitochondria contain unsaturated fatty acids, iron sulphur clusters, densely packed proteins, and multiple copies of mitochondrial DNA (mtDNA), all of which are essential to the mitochondrial function and typical targets of oxidative damage [119]. The most vulnerable to oxidative damage, due to their proximity to ROS production, are the ETC complexes, which include cardiolipin [120]. Moreover, oxidative stress triggers the peroxidation of cardiolipin, thereby impairing cristae formation [121]. On the other hand, the respirasome and the detachment of cytochrome c, which is a mobile electron carrier in the inner mitochondrial membrane (IMM), are also affected by the peroxidation of cardiolipin [122]. Mitochondrial DNA is associated with the IMM and vulnerable to oxidative damage due to the lack of protective histones. Furthermore, damage to mtDNA results in ATP synthesis reduction, reduced ETC activity and a further increase in electron slippage to oxygen, thus setting up a feed-forward cycle of ROS-induced ROS production [123,124,125]. Hence, at the centre of an efficient cardio-protection strategy, in the early stages of HF, is the preservation of the functionality of the mitochondria in cardiomyocytes, which represents a consistent challenge.

### 2.2. Endoplasmic Reticulum Stress in HF

As previously shown for the role of mitochondrial dysfunction, stressful conditions occurring at the level of the endoplasmic reticulum (ER) stress also seem to play a crucial role during the early phases of HF [126,127,128]. In particular, there is evidence that ER impairment is accompanied by enhanced oxidative stress which, in turn, is connected to various cardiovascular conditions, such as cardiac hypertrophy and, at the late stages, HF [129,130].

ER and mitochondria are capable of regulating oxidative stress. In particular, in models of heavy-metal-induced myocardial dysfunction, an increase in the level of stress markers of ER occurs. This implies that oxidative stress and heart dysfunction are connected to extreme ER stress. Moreover, data have shown that when induced by excessive ER stress, protein damage and intracellular calcium ion (Ca^2+^) anomalies trigger heart dysfunction [130,131]. Several toxicants potentially contribute to heart injury via this mechanism. The effect of these toxicants can be counteracted by natural antioxidants which efficiently revive ER stress levels in the heart. Hence, by regulating the ER-related pathway, natural antioxidants can exert a protective role against cardiotoxic agents via a significant reduction in ER stress. The mechanism of cardioprotection obtained by means of natural antioxidant supplementation can possibly be explained by regulation of the ER-related pathway. In particular, it has been found that ER and oxidative stress lead to impairment of the so-called unfolded protein response (UPR), which is a compensatory mechanism that contributes to maintenance of the myocardial cell integrity. This involves complex molecular pathways, probably driven by eukaryotic translation initiation factor 2 alpha (eIF2α) phosphorylation and protein kinase RNA-like ER Kinase (PERK) and leading to a compensatory response to ER stress, although this needs to be better clarified [130,131,132,133,134,135,136]. Furthermore, an increase in intracellular Glucose-Regulated Protein 78 (GRP78), considered to be a chaperone of ER stress and, most importantly, a regulator of UPR, has been shown to play a role in these mechanisms [132,133,134,135,136].

Therefore, it is likely that active ingredients of plant extracts, once given at an appropriate dosage and formulated to obtain the due concentration at the active site of action, may produce a beneficial effect to counteract cardiomyocyte ER stress.

### 2.3. Imbalanced Metalloproteinase Regulation in HF

The extracellular cardiac matrix (ECM), which is a complex architectural network, maintains a balance between the degradation and deposition of matrix proteins by preserving the correct cardiac geometry whilst also allowing the myocardium to maintain its structural integrity [137]. The ECM aids in the proper functioning of heart cells, forming a specific scaffold which allows for the anchoring of these proteins [138]. A range of physiological and pathological processes, such as cell proliferation and differentiation, and tissue morphogenesis, are influenced by the ECM turnover [139]. The ECM is also responsible for the transduction of mechanical forces within the cardiac vessels and heart, and for promoting diastolic compliance in the arterial wall.

Matrix metalloproteinases (MMPs) are enzymes capable of degrading both of the ECM structural proteins being modulated by metalloproteinase endogenous tissue inhibitors (TIMPs). These are the fundamental mediators of ECM remodelling. In normal circumstances, these processes are highly regulated. The roles of MMPs and TIMPs can be both positive and negative in cardiac remodelling [138,139], with TIMPs controlling MMPs, minimising the degradation of the matrix. The MMPs play a pathological and irreversible role in remodelling ECM, which is important in both compensatory cardiac hypertrophy and acute decompensated heart failure [140]. MMPs also mediate the ventricular remodelling caused by myocardial infarction or viral myocarditis [140]. Several studies indicate that the serum level of metalloproteinase-2 (MMP-2) is an autonomous predictor of mortality in HF patients [141]. MMPs are made up of five main groups, including gelatinases, collagenases, matrilysins, stromelysins, and metalloproteinases of membrane-type/membrane-type MMPs. The divisions of these groups are based upon their structure and the specificity of the substrate [142]. Two of these—Gelatinase A and Gelatinase B, (also known as MMP-2 and MMP-9, respectively)—are directly involved in the pathogenesis of coronary thrombosis, atherosclerosis, myocardial infarction, and heart failure [143]. Furthermore, MMP-2 causes migration of the cell, growth, differentiation, and inflammation [144]. These enzymes cleave structural proteins from the elastin and collagen networks. The known spectrum of substrates for MMP-2 not only contains the components of extracellular matrixes, such as collagens or elastin, but MMP-2 is also able to digest parts of the contractile apparatus, such as the myosin 1 light chain or troponin I [143]. It has been proven that, during ischaemia, the proteolytic function of MMPs in myocardium generally increases. Moreover, alterations in the balance of MMPs and TIMPs could contribute to acute myocardial ischaemia-reperfusion injury [145]. Conversely, MMP-2 is associated with myocardial dysfunction, contributing to the development of cardiomyopathies [146,147,148,149,150]. The occurrence of an imbalanced regulation of the NT-truncated intracellular isoform of MMP-2 (NT-MMP-2) as a consequence of mitochondrial dysfunction and oxidative stress seems to be involved in myocardial injury [151,152]. Due to the strong connection between NT-MMP and the mitochondria, it is likely that pathophysiological events, subsequent to hyperglycaemia, may underlie a cascade of events mediated by oxidative stress which are associated with mitochondrial dysfunction [152,153,154]. These events cause an altered modulation of MMPs and, at the end stage, lead to failing myocardium, such as that which can be seen in the hearts of diabetic patients.

Recently, we examined an exaggerated production of superoxide anion (NADPH oxidase-dependent), which occurs in hyperglycaemic rats, gradually impairing the function and myocardial structure [155]. In particular, we found that during the initial stage of cardiac injury brought on by chronic hyperglycaemia, the reduced fractional shortening and ejection fractioning are associated with biochemical changes, suggesting the involvement of mitochondrial functionality. In particular, in a model of streptozotocin (STZ)-induced hyperglycaemia, we found that increased ROS production is accompanied by an increased expression of the mitochondrial translocator protein (TSPO): a protein located at the level of the outer mitochondrial membrane [154,155,156] which has been found to be imbalanced in post-ischemic cardiac damage [157]. Moreover, STZ-induced hyperglycaemia is associated with an increased expression of selective Voltage-Dependent Anion Channel-1 (VDAC1), which contributes, alongside TSPO, to the regulation of calcium traffic at the level of the transition pore of the mitochondrial membrane. Under these experimental conditions, MMP-2 was downregulated and NT-MMP2 was found to translocate at the level of the mitochondrial membrane, thereby suggesting that these mechanisms could contribute to functional changes seen in the heart of hyperglycaemic rats. These effects may be counteracted by natural antioxidants (see below).

### 2.4. The cGMO/NO Pathway in HF

There is evidence that, through important roles in the cardiac myocyte, nitric oxide (NO), nitric oxide synthase (NOS), and soluble guanylate cyclase (sGC) all oppose pathological remodelling when functioning normally [158]. A large body of work supports that in HF and in conditions predisposing to HF, NO, which is the activator of soluble guanylate cyclase, also becomes dysregulated [159]. The cardiovascular tissue release of NO can be directly reduced by ROS in the absence of effects on NOS expression and activity [159]. Correspondingly, human studies have shown that there is a predisposition to HF in conditions associated with reduced endothelial NO release. These reduced endothelial NO release conditions include ageing, diabetes, and obesity [160]. In fact, based on these human data, a current proposed model postulates that the pathogenesis of HFpEF is driven by the generation of reduced NO-induced cyclic guanosine monophosphate (cGMP) [161,162]. Animal models have also shown that endothelial nitric oxide synthase (eNOS) becomes uncoupled not only in hypertension [163], but also in a pressure overload stress model [164]. This promotes the production of pathologic reactive oxygen species (ROS), rather than NO. Additionally, under conditions of oxidative stress and multiple HF risk factors, soluble guanylate cyclase can be directly modified and rendered unable to generate cGMP in response to NO. In fact, in a pressure-overload stress model, it has been reported that the translocation of sGC is involved in sGC oxidation and contributes to depressed NO-stimulated sGC activity [164].

Moreover, the failing heart not only exhibits changes in cGMP catalysis through the upregulation of cGMP-specific phosphodiesterases (PDEs), but also alterations in sGC augmentation. Part of an 11-member family, PDEs (PDE1 to PDE11) catabolise cAMP, cGMP or both, depending on the specific PDE [165]. Recently, PDE5 and PDE9, both of which are cGMP-specific PDEs, have been studied. According to samples taken from end-stage failing myocardium, PDE5 expression increased in the failing human left ventricular (LV) [166]. Furthermore, in the LV of mice subjected to experimental transverse aortic constriction (TAC), aside from protein expression, the total cGMP phosphodiesterase activity increased [167]. Studies on pharmacological and genetic manipulation in mice have also confirmed the pro-remodelling role of the enzyme in the cardiomyopathies and the effectiveness of PDE5 inhibition in opposing cardiac remodelling [165]. More recently, in the failing human LV, an increase in the cGMP-selective PDE9 was observed in both Reduced as well as Preserved Ejection Fraction Heart Failure (HFrEF and HFpEF, respectively). This was observed in the LV of mice subjected to TAC as well. After thoracic or abdominal aortic constriction, as well as in response to isoproterenol infusion, genetic deletion or pharmacological inhibition of PDE9, LV remodelling improved [168,169].

The aforementioned observations describe a general model in which alterations in components of both cGMP-generating and anti-cGMP-generating components lead to a net reduction in myocardial cGMP, subsequent cardiac remodelling and failure. Studies on the myocardial tissue of HFpEF patients, compared with that of not-failing tissue patients with aortic stenosis or with HFrEF tissue, have demonstrated a reduced cGMP concentration [170]. Reduced myocardial cGMP has also been observed in a rat model of HFpEF [171], and tissue from patients display increased PDE9A and cGMP-esterase activity [170]. On the other hand, in the presence of PDE5 and PDE9 inhibition, numerous animal studies have also identified a net increase in cGMP [168,169]. Therefore, the rationale behind pharmacological and nutraceutical strategies to augment intracellular cGMP in patients with HF is supported by animal studies of cGMP regulating molecules and direct observations in humans with HF.

### 2.5. HF and Sodium-Glucose Cotransporters

Evidence has been provided demonstrating that sodium–glucose cotransporter (SGLT2) inhibitors may produce benefits in cardio-renal dysfunction in diabetic patients. This has been assessed by means of an extensive review by Verma and McMurray reporting large-scale clinical trials of type 2 diabetes mellitus (T2DM) patients with either multiple cardiovascular risk factors or established cardiovascular disease [172].

In particular, it has been reported that many of the potential benefits of regulating SGLT2 in both diabetic and non-diabetic patients with HF have been attributed to the slowing of the atherothrombotic processes which accompany type 2 diabetes mellitus (T2DM), and the improvement in the cardio-renal consequences of T2DM, which can lead to myocardium decay. Moreover, recent evidence suggests that SGLT2 modulation is accompanied by additional effects which may be beneficial to HF management. In particular, it has been postulated that SGLT2 inhibitors may improve and/or optimise cardiac energy metabolism.

These agents may provide improvement in the cardiac output and cardiac efficiency by acting on the substrate efficiency and myocardial energetics [172]. Furthermore, in the presence of T2DM and/or HF, it has been suggested that the metabolic flexibility of the heart is impaired as it relates to substrate utilisation [173].

A build-up of free fatty intermediates may result from an over-reliance on non-esterified fatty acids (NEFAs) as a substrate for ATP generation. This action may also promote the development of diastolic dysfunction and lipotoxicity [174].

In those with diabetes, SGLT2 inhibitors may offer an optimal alternative myocardial fuel source [175]. This alternative fuel source is more cost-effective in those with T2DM, as SGLT2 inhibitors are known to promote ketone body β-hydroxybutyrate (βOHB) production [176]. It has been postulated that elevated ketone levels may be the result of an effort to increase glucagon levels through a possible decrease in βOHB excretion via the kidneys. The concept underlying this is that ketone-body βOHB is a “super fuel”, oxidised by the heart instead of glucose and NEFAs. In the failing heart, ketones may improve the cardiac function and increase the mechanical efficiency [177,178]. This is an interesting postulate, although cogent data supporting this thesis are scarce. Preliminary studies carried out in pigs following myocardial infarction showed that empagliflozin increases the consumption of myocardial ketone while simultaneously reducing the production of lactate, as well as the consumption of cardiac glucose [179]. Another hypothesis is that this is due to the inhibition of histone deacetylase, promoting increases in the levels of SGLT2-inhibitor-induced βOHB and preventing pro-hypertrophic transcription pathways [180].

It is also possible that a decrease in βOHB oxidation promotes a decrease in acetyl-CoA, which is a derivative of ketone oxidation. These decreased levels, in turn, increase glucose-derived pyruvate oxidation and thus improve myocardial glucose metabolism. A decrease in the supply of acetyl-CoA may result in an improved production of mitochondrial energy and, consequently, in a decrease in the harmful hyperacetylation, thus aiding the mitochondrial enzymes [180].

It has been postulated, using an untargeted metabolomic approach, that the inhibition of SGLT2 promotes the degradation of branched-chain amino acid (BCAA), which may provide an alternative source of fuel to the failing myocardium. In HF patients, the impaired degradation of BCAA in myocardium may contribute to aberrant bioenergetics [181]. As intriguing as these findings are, it must be noted that we currently lack definitive evidence that links the beneficial effects of SGLT2 inhibition to myocardial energetics.

A promising emergent hypothesis is that direct inhibition of the Na+/H+ exchanger (NHE) 1 isoform in the myocardium may be an effect of SGLT2 inhibition and consequential effects on Na+/H+ exchange in the myocardium [182,183].

In experimental models of HF, it has been demonstrated that NHE1 activation results in increased cytosolic sodium and calcium. Recently, Ulthman et al. demonstrated that the cardiomyocyte NHE is inhibited by the SGLT2 inhibitor empagliflozin, thereby increasing mitochondrial calcium levels while reducing the levels of cytoplasmic sodium and calcium [183].

In the heart, the mechanism by which these effects occur on cardiomyocyte NHE remains elusive, as SGLT2 receptors are not expressed. Notably, it has been suggested that natriuresisis is promoted in the proximal tubule by SGLT2 inhibitors, which downregulate the NHE3 activity [184]. In HF patients, the expression of NHE3 is increased and known to mediate the reuptake of tubular sodium. As an additional mechanism, the inhibitory effects on NHE3 may serve to reduce cardiac failure and restore whole-body sodium homeostasis. Hence, prevention and/or treatment of HF may be possible by these agents through a common cardio-renal mechanism, such as the inhibition of NHE1 and NHE3 [182].

Cardiac fibrosis and SGLT2 inhibition are widely regarded as common final pathways through which HF develops. Cardiac structural remodelling is universally involved in this mechanism due to the deposition of ECM proteins by cardiac fibroblasts, which accelerate the development of HF by impeding ventricular compliance [185]. Recent experimental data on myocardial infarction showed the significant cardiac anti-fibrotic effects of dapagliflozin on the suppression of collagen synthesis, an effect made possible by the increased activation of M2 macrophages and inhibition of myofibroblast differentiation [186]. Additionally, preliminary studies, using human cardiac fibroblasts measured by the collagen fibre alignment index, have demonstrated that empagliflozin significantly attenuates the activation of TGF-β1-induced fibroblasts while reducing cell-mediated ECM remodelling [187]. Moreover, evidence demonstrates that empagliflozin suppresses the expression of multiple key pro-fibrotic markers, including type I collagen, α-smooth muscle actin, the connective tissue growth factor, and MMP-2 [187]. Therefore, an emerging postulate is that the inhibition of SGLT2 may have direct and favourable effects on one of the most important factors of HF—the cardiac fibroblast phenotype and function. These effects indicate potential for using nutraceutical supplementation with natural SGLT2 inhibitors, such as a phlorizin-rich apple extract.

### 2.6. Impairment and Senescence of Cardiac Stem Cells in HF

It is known that myocardial infarction and ischaemic heart disease are the major determinants for the development of HF [188,189]. Furthermore, where HF is of a nonischaemic origin, in cases of structural and “functional” cardiomyopathies, the lack of myocardium is the primary issue to be addressed for a robust cardiomyocyte replacement [190,191]. Regenerative medicine aims to find a therapy that is both effective and broadly available to refresh the contractile muscle cells lost and/or rendered permanently dysfunctional as a consequence of the primary injury [190,191]. Unfortunately, the predominant scepticism about the intrinsic endogenous regenerative capacity of the adult mammalian heart, including the human heart, has produced often contradictory approaches in myocardial repair/regeneration [191]. This skepticism can only be overcome if hard, clean and clear scientific data are obtained, thereby eliminating the need for interpretations and opinions. It is unlikely that any clinical repair or regeneration protocol will ever be able to answer the question regarding the feasibility of functionally regenerating the failing human heart [191]. It is well-known that the mammalian heart, including the human heart, contains a pool of resident tissue-specific cardiac stem/progenitor cells—the endogenous Cardiac Stem Cells (CSCs)—referred to as eCSCs when in the myocardium, and CSCs when isolated and studied in vitro [192,193,194]. The eCSCs have been identified as a small cardiac cell population through the expression of specific membrane markers, in particular, the stem cell factor (SCF) receptor kinase c-Kit [195], Sca-1 [196], and MDR-1 [197]. In vitro and in vivo experiments have clearly shown that CSCs, being multipotent, clonogenic, and self-renewing, have all the characteristics expected of a tissue-specific stem cell. Differentiation of the main myocardial cell is possible in vivo and in vitro [192]. Several studies have reproduced the findings that while the exhibition of c-Kit cell dysfunction is present in W locus mouse mutants (W/Wv) [198], c-Kit signalling in vitro promotes the survival, growth, and proliferation of human CPCs [199]. Indeed, W/Wv mice display impaired cardiac recovery after infarction [200], diminished cardiac function with advanced age [201], and compromised c-Kit cell differentiation into cardiomyocytes [202]. Blunted reparative responses were exhibited in both myocardial injuries in bone marrow c-Kit^pos^ cells from W locus mutants and in vitro cells with silenced c-kit [198].

Furthermore, deletion of the c-Kit gene, as it occurs in homozygous W-mutated mice [200], causes premature murine death, because c-Kit gene deletion is incompatible with life. However, during embryonic life, c-Kit-defective adult mouse hearts appear to develop normally [203], while adult myocardial infarction model c-Kit Cre-KI mice have demonstrated a significant defect in the regeneration potential [204]. Therefore, it appears that the roles that c-Kit plays in cardiac regeneration are different to those it plays in heart formation/development. This suggests that the molecular mechanism underlying cardiac regeneration differs from that of cardiac generation. Despite all of the attempts that are currently ongoing to decode the pathways of developmental cardiac generation and neonatal heart regeneration to instruct effective protocols of adult cardiac regeneration, cardiac generation has still not been predicted [205]. Finally, the role of c-Kit in cardiac pathology was evaluated in several models, such as aging cardiomyopathy [206,207], doxorubicin-induced cardiomyopathy [208], and chronic heart failure [209,210]. In particular, Huang et al. developed a “paediatric” model of doxorubicin-induced cardiotoxicity, in which juvenile mice were exposed to doxorubicin (DOXO) using a cumulative dose that did not induce acute cardiotoxicity [211]. These juvenile mice developed normally and had no obvious cardiac abnormalities as adults. However, these hearts did have reduced numbers of c-Kit^pos^ cardiac cells and abnormal vasculature, which correlated with an increased sensitivity to stimuli, both physiological and pathological. When subjected to myocardial infarction, adult mice developed a more pronounced cardiac decompensation, which correlated with a failure to increase the capillary density in the injured area. It was subsequently demonstrated that DOXO-induced cardiomyopathy is due to a depletion of the functional c-Kit^pos^ CSC pool, and can be rescued by restoring its function via nutraceutical supplementation [212].

## 3. Candidates for Nutraceutical Supplementation in the Failing Myocardium according to the Aforementioned Emerging Mechanisms

### 3.1. Coenzyme Q10

Coenzyme Q10 (CoQ10), also defined as ubiquinone (in the oxidised form) or ubiquinol (in the reduced form), plays a key role in the functioning of the MC. CoQ10 is responsible for the transfer of electrons from complex I and II to complex III, thus promoting ATP generation [213] (Figure 1). Moreover, CoQ10 is involved in the so-called proton motive Q-cycle, which enables the passage of protons across the internal mitochondrial membrane [214]. Q10 has a highly lipophilic molecular structure and is related to vitamin K, where “Q” represents quinone and “10” refers to the 10-isoprene group. [215]. CoQ10 is ubiquitous in most mammalian tissues, with particularly high levels in organs, such as the heart, with the highest rate of metabolism [215]. The higher concentration of CoQ10, compared to other carriers, balances its slower cyclic oxidation/reduction rate [30]. Therefore, CoQ10 deficiency in the cardiac mitochondria leads to the dysfunction of mitochondrial respiration. Conversely, CoQ10 supplementation could improve the mitochondrial function [18]. Preclinical data suggest that CoQ10 has important anti-inflammatory properties and is able to protect endothelial function from damage [19]. In fact, CoQ10 also regulates eNOS function in different cellular membranes. Therefore, the depletion of CoQ10 can promote the uncoupling of eNOS, making it an additional source of ROS and thereby shifting the nitrous–redox balance towards oxidation [20].

Interestingly, data have demonstrated a protective role of CoQ10 in high glucose-induced endothelial progenitor cell (EPC) dysfunction [31]. In particular, the administration of CoQ10 reduces apoptosis cell death and increases the mitochondrial membrane potential. In addition, CoQ10 is able to reduce ROS production, enhancing eNOS/Akt activity and upregulating HO-1 expression [31] (Figure 2).

Physiologically, about 50% of CoQ10 is ingested, while the remaining 50% is endogenously synthesised through the mevalonate pathway, which is inhibited by a statin action [32]. Indeed, in isoproterenol-induced HF in rats, Garjani et al. showed a severe LV dysfunction when the animals were treated with a high dose of atorvastatin. Conversely, it has also been reported that the co-administration of CoQ10 with a lower dose of atorvastatin ameliorates myocardial necrosis and fibrosis and the hemodynamic depression, improving left ventricular (LV) dysfunction [33]. Hence, patients with a cardiovascular risk and/or disease treated with statins may experience CoQ10 deficiency [18]. In patients with HF, the severity of disease is correlated with CoQ10 deficiency [34]. Experimental evidence suggests that lipid peroxidation, resulting from oxidative stress, causes a reduction in CoQ10 levels [35]. At first, in a preliminary clinical study conducted in patients with HF, CoQ10 was defined as an independent predictor of mortality [36]. However, this was not confirmed by the data obtained from the larger CORONA study.

According to the results of the CORONA study, the treatment of patients with rosuvastatin failed to reduce nonfatal myocardial infarction, nonfatal stroke due to ischaemic cardiomyopathy, and the primary endpoint of death from cardiovascular causes [37]. However, the treatment of patients with rosuvastatin did reduce CoQ10 concentrations. Nevertheless, the administration of rosuvastatin in patients with a low baseline of CoQ10 did not lead to a worse outcome [37]. In accordance with this, CoQ10 deficiency was not considered an independent prognostic factor in HF [37]. In light of this evidence, CoQ10 deficit could play a causal role in patients suffering from HF, even those not treated with statins. CoQ10, as an established mitochondrial therapy, ameliorates CoQ10 biosynthesis defects through the constant dosing of CoQ10 dietary supplementation [38]. However, despite the lack of firm evidence of its benefit in cardiovascular diseases, CoQ10 supplementation has been studied for several decades. Though several clinical trials have shown that CoQ10 may enhance Left Ventricular Ejection Fraction (LVEF), other well-designed studies are needed to assess its effect on patient outcome [21]. Recently, in the Q-SYMBIO trial, the effects of CoQ10 were studied in 420 patients with systolic HF. The results showed a significant amelioration in symptoms and a substantial decrease in major adverse cardiovascular events [21]. However, the study was underpowered and failed to provide enough evidence of a benefit in this population. Therefore, it is still unclear whether CoQ10 administration benefits the outcome and symptoms in HF [21,22]. Moreover, in the recent guidelines on the treatment of HF, there is no mention of CoQ10 [19].

### 3.2. Bergamot Polyphenols

Evidence has been collected showing that bergamot fruit (both juice and albedo) produce beneficial effects in the cardiovascular system [24,39,40,41,42,43,44]. In fact, being rich in polyphenols, bergamot derivatives produce an antioxidant response both in vitro and in vivo, leading to a reduction in cholesterol, glucose and triglyceride serum levels, which is an effect accompanied by a reduction in systemic inflammation and, subsequently, an improvement in the endothelial function. These effects were confirmed in patients, mostly by means of a bergamot polyphenolic fraction (BPF) extract, which displayed a potent effect in modulating the liver traffic of lipoproteins and was able to counteract Non-Alcoholic Liver Disease (NAFLD). The major components of BPF are naringin, neoeriocitrin and neohesperidin and glycosylated polyphenols, such as bruteridin and melitidin, which have been found to produce the inhibition of 3-hydroxy-3-methyl-glutaryl Coenzyme A (HMGCoA) reductase [23,25,45,46].

Recent data published by Carresi et al. have also shown that BPF treatment leads to relevant benefits in doxorubicin (DOXO)-induced cardiac damage in rats, preserving LV contractility and attenuating pathologic myocardial remodelling [212]. Intriguingly, our data show that BPF strongly prevented the induction of ROS, the excessive expression of pro-autophagic mediators, and myocyte apoptotic cell death. Moreover, there is evidence that BPF is able to attenuate attrition in cardiac-resident stem cells (eCSCs), promoting newly formed myocytes in DOXO-treated rats [212].

In fact, Carresi et al. showed, for the first time, that BPF is also able to attenuate attrition in endogenous cardiac stem cells (eCSCs), thereby improving the number of resident c- Kit^pos^CD45^neg^CD31^neg^ eCSCs. On the other hand, BPF inhibited 8-hydroxy-2′-deoxyguanosine (8-OHdG) nuclear accumulation, and promoted the replenishment of cardiomyocytes with an increased number of small, newly formed BrdU^pos^ myocytes after DOXO administration [212] (Figure 2). This fits with previous experimental evidence correlating DOXO-induced cardiac damage with stem cell impairment. Indeed, Burridge et al. showed, in patients with breast cancer undergoing DOXO-induced cardiotoxicity, that human-induced pluripotent stem-cell-derived cardiomyocytes are compromised by atrcaciclyne treatment [216]. In addition, it has recently been reported that the intravenous administration of cardiac progenitor cell-derived exosomes protects against doxorubicin/trastuzumab-induced cardiac toxicity [217].

Therefore, the main findings from the Carresi et al. study revealed the highly beneficial cardio-protective effects of BPF against DOXO-induced cardiomyopathy through its direct scavenging and antioxidant properties.

The antioxidant effect of BPF enables it to interfere with the production of DOXO-induced free radicals, excessive autophagy activation, and apoptosis, thus preventing the compensatory mechanisms and pathological changes that lead to the development of cardiomyopathy.

Overall, these data show a widely beneficial effect, highlighted by BPF, not only on cardiomyocytes, but also likely on the population of endogenous cardiac stem cells (eCSCs). These findings likely represent only part of the cardio-protective effects of bergamot-derived polyphenols.

BPF has clear protective effects on the heart and is likely involved in the maintenance of functional endogenous cardiac stem cells. Even if some of the protective effects of BPF can be clearly attributed to its direct scavenging and antioxidant properties, the protective molecular mechanism involved in the maintenance of resident endogenous cardiac stem cells is not yet fully understood.

Further work is required to understand the specific molecular and genetic mechanisms underlying the onset of cardiomyopathy induced by DOXO and to confirm our findings on the role of BPF in cardio-protection.

### 3.3. Olea Europea L. Extract

Several studies credit the Olea Europaea leaf extract (OLEX) for the majority of the beneficial effects that the Mediterranean diet has on human health [218,219,220,221,222]. Initially, the richness of monounsaturated fatty acids (MUFA), and in particular that of oleic acid, was considered the major healthful characteristic of OLEX. Following other observations made for various aliments rich in MUFA, such as sunflowers, rapeseeds, and soybean, it is quite clear that none of those aliments are comparable to OLEX as healthy food [223,224], even when taking the role of some “minor components” into consideration. The compounds found in OLEX, when consumed in crude form, are able to maintain their biological activity. In the unsaponifiable fraction of olive oil, there are more than 200 “minor components”. These minor components represent about 2% of the total weight and include a number of heterogeneous compounds that are chemically unrelated to fatty acids [225,226]. Recently, the nutraceutical properties of OLEX compounds and the antioxidant activity provided by these compounds have been the focus of a great amount of attention. In OLEX, the most abundant antioxidants are lipophilic and hydrophilic phenols [227], which are normally synthesised by the plant to react to various insect injuries and/or pathogen attacks [228,229].

Nutraceutical properties have mainly been attributed to secoiridoid oleuropein (OL) and its derivatives; the main alcohol 3, 4-dihydroxyphenyl ethanol, also known as hydroxytyrosol (HT); and *p*-hydroxyphenyl ethanol or tyrosol [48,230]. These compounds are released during the extraction process from the olive drupes into OLEX. In particular, OL is abundant in unprocessed olive leaves and drupes, while a higher concentration of HT may be found in the drupes and in olive oil, due to chemical and enzymatic reactions that occur in the plant during maturation of the fruit [49,50]. In addition, many agronomic factors may influence the final concentration in OLEX, such as the geographic origin of olive drupes, cultivar, olive tree irrigation, and ripening stage, as well as the various oil extraction conditions during crushing, malaxation, and OLEX separation [50].

The in vivo antioxidant activity of OL and HT is related to their high levels of bioavailability [51,52]. Various studies have documented a high degree of absorption of both compounds, which is fundamental to exerting their metabolic and pharmacokinetic properties [48]. OL and HT behave as antioxidants acting as anti-oxygen radicals, free radical scavengers, radical chain-breakers, and metal chelators. They can break peroxidative chain reactions and scavenge the peroxyl radicals with their catecholic structure, producing stable structures [53]. In vitro studies have shown a decrease in ROS production after treatment with OL or HT, suggesting a chelating action on such metal derived from copper-ion-induced oxidised low-density lipoprotein (oxLDL) [54]. Furthermore, in rats treated with doxorubicin (DOXO), it has been observed that OL could prevent cardiomyopathy [55]. In addition, Granados et al. have reported that in rats with breast cancer, HT attenuated DOXO-associated chronic cardiac toxicity, thus ameliorating mitochondrial dysfunction [56]. In particular, HT treatment significantly prevents mitochondrial swelling and vacuolization, improving the integrity of complex-III of the Mitochondrial Electron Transport Chain (METC) after DOXO treatment [56] (Figure 1). Treatment with OL also reduced the infarct size in normal and hypercholesterolemic rabbits [57]. Additionally, OL protection was associated with a reduction in total cholesterol and triglyceride levels in reperfused myocardium [57]. Furthermore, HT reduced the protein expression related to ageing, the infarct size, and apoptosis in cardiomyocytes [58]. In tyrosol-treated rats compared to non-treated animals, a reduced infarct size was observed, with a concomitant improvement in the myocardial function [59]. In particular, OL was able to inhibit the myocardial infarction size and reduce the levels of Creatine-Kinase MB isohenzyme (CK-MB) and lactate dehydrogenase (LDH) in a model of myocardial ischemia-reperfusion in rats. The mechanism of olea derivatives is still unknown. However, recent data showed that, in myocardial I/R rats, OL modulated the ERK pathway and suppressed the induction of p53, p-MEK, and p-ERK protein expression [60]. Cisplatin-induced acute renal injury was attenuated by OL through the inhibition of p53 and ERK signalling in mice [61]. Moreover, evidence exists that the cardioprotective effects of OLEX and HT are mainly exerted via endoplasmic reticulum (ER) stress prevention [62]. Indeed, the administration of HT prevents ER-stress-induced apoptosis by inhibiting the mRNA and protein expression of GRP78 and CHOP in hypoxia-induced H9c2 cells. On the other hand, pre-treatment with OLEX prevents inflammation and reduces the infarct size, improving ejection fraction and shortening fraction FS in isoproterenol-induced myocardial infarction in rats [62]. Finally, recent data suggested that OL prevents oxidative stress directly scavenging O_2_^-^ in AngII-mediated human vascular progenitor cell (VPC) depletion. Furthermore, evidence has been collected suggesting that OL exerts its protective effects via upregulation of the ERK1/2-Prdx1 and 2 and Akt/eNOS signalling pathway [63] (Figure 2). Therefore, derivatives of *Olea Europea* L. appear to represent natural compounds able to exert a cardioprotective response acting on pathways which lead to the impairment of cardiomyocytes.

### 3.4. Apple-Derived Natural SGLT_2_-Inhibitors

The principle phenolic-glucoside in apple trees is phlorizin, which is present in roots, bark, shoots, and leaves [231]. It is a naturally competitive inhibitor of SGLT2, providing the first insights into the potential efficacy of this compound [64]. Many effects of the phlorizin-rich extract contribute to its potential use in treatments to ameliorate diabetes and other metabolic disorders. In fact, it has been reported that apple juice and apple extracts contain a total phenolic concentration of 11%–36% phlorizin, which could inhibit oxLDL levels [65]. Additionally, in isolated coronary artery rings, the aglycon of phlorizin—phloretin—produces endothelium-independent relaxation [66,67].

Many pre-clinical studies have found that phlorizin produces effects such as lowering serum glucose and improving insulin-resistance [68,69,70]. In addition, with the scope of observing the absorption of delayed intestinal glucose, in 1997, Japanese researchers demonstrated an effect on the post-prandial rise of serum glucose after orally administering phlorizin to mice [71]. Following these observations, investigators at Tanabe Seiyaku Co., a Japanese pharmaceutical company, developed a phlorizin derivative; 3-(Benzo[b]furan-5-yl)-2′,6′-dihydroxy-4′-methyl-propiophenone-2′-O-(6-O-methoxycarbonyl)-β-D-glucopyranoside (T-1095) [72]. In particular, it was demonstrated that the oral administration of T1095 leads to an improvement in hyperglycaemia and insulin resistance in the skeletal muscle of streptozotocin (STZ)-induced diabetic rats [72]. In addition, data have also been provided showing that T-1095 improves hyperglycaemia by suppressing the renal reabsorption of glucose, possibly via inhibition of the expression of Na+-glucose cotransporters (SGLTs) and of the abnormal expression of GLUT2 in the kidney [73]. Therefore, due to the emerging evidence on the successful use of SGLT2 inhibitors in diabetic cardiomyopathy (see Section 2.5), it is likely that the phlorizin-rich apple extract may represent a potential candidate for supplementation in diabetes-associated impairment of the cardiac performance.

### 3.5. Grape Seed Extract

The grape seed extract (GSE) represents a source of active ingredients which have been suggested to have beneficial effects on many disease states, including cardiovascular disorders; for a review, see [232]. Among the active ingredients found in GSE, proanthocyanidins (PCs) were found to be able to produce therapeutic, biological, pharmacological and chemoprotective properties against free radicals and oxidative stress see [232]. Interestingly, grape seed PCs have shown cardioprotective effects through the modulation of mitochondrial and lysosomal function [74]. In fact, in isoproterenol- treated rats, co-treatment with PCs reduced the activities of lysosomal enzymes in the heart tissues and increased respiratory chain enzymes and mitochondrial activity [74].

Many of the cardiovascular benefits attributed to GSE rich in PCs have been found to be related to their antioxidant profile. Indeed, GSE exhibited dramatic concentration-dependent antioxidant activity [75]. Additionally, in mice, the dose-dependent protective ability of GSE was excellent against 12-O-tetradecanoylphorbol-13-acetate (TPA)-induced hepatic peroxidation, as well as DNA fragmentation, peritoneal macrophage activation, and brain lipid peroxidation [76]. Moreover, in a primary culture of human oral keratinocytes, GSPE demonstrated significant protection against DNA damage, oxidative stress induced by smokeless tobacco, and apoptotic cell death [77]. Additionally, GSE was shown to enhance the cytotoxic effects of idarubicin and 4-hydroxyperoxycyclophosphamide on Chang liver cells [78] and exerted a direct cytotoxic effect on cultured human cancer cells [79]. Finally, Bagchi [80] and Ray [81] reported that GSE also leads to significant protection against chemotherapeutic drug-induced cytotoxicity in common liver cells via the counteraction of both programmed and non-programmed cell death.

This was confirmed in cardiomyocytes [82]. Indeed, Sato et al. showed that GSE rich in PCs reduced the apoptotic cell death of cardiomyocytes in ischemia/reperfusion-induced damage, representing an effect driven by the activation of jnk-1 and c-jun [82]. Therefore, GSE rich in PCs leads to an antioxidant and anti-apoptotic effect in failing myocardial cells by means of antagonistic properties on ischaemia–reperfusion-induced activation of c-jun and by modulating the pro-apoptotic gene JNK-1.

## 4. Conclusions and Future Perspectives

The use of nutritional support and nutraceutical supplementation in HF patients has been suggested, over the last decades, on the basis of both pre-clinical and clinical studies carried out worldwide. However, there are consistent controversies with regards to defining the real contribution of nutraceutical interventions in the course of the disease and their correct impact on approaching hospitalisation and death in patients undergoing HF. Recent studies exploring pathophysiological mechanisms characterizing the early stages of HF, mostly in the course of cardiometabolic disorders underlying impaired myocardial dysfunction, have allowed the possible identification of a condition of “cardiomyocyte frailty” in which oxidative stress and mitochondrial dysfunction seem to play a crucial role. Under these conditions, protective mechanisms which include autophagic responses and endogenous anti-oxidant enzyme overexpression are activated to antagonise the apoptotic cell death of myocardial cells. This stage of the disease is accompanied by an alteration in ER, calcium imbalance and an inappropriate regulation of MMPs, which have been shown to increase the stiffness of the left ventricle, leading to diastolic dysfunction. At this stage of the disease, based on their antioxidant and anti-apoptotic properties, several plant extracts, characterised by the specific modulation of oxidative processes which should occur at the early stages of cardiac dysfunction, are expected to play a role in counteracting the development of HF. To this end, contribution to the mobilisation of stem cells could have a further beneficial effect in supporting the failing myocardium at any stage of the disease. On the basis of this preliminary evidence, it is likely that a due combination of plant extracts, well-characterised in terms of active ingredients, may represent a potential resource to support endogenous compensatory mechanisms which characterise the “frail myocardium” at the early stages in the development of HF. This should be defined by means of multicentric clinical studies, randomised, double-blind and placebo-controlled, to be carried out on a large number of subjects, in order to verify the efficacy and safety profile of nutraceuticals. An assessment of their placement in therapy according to the stages of the disease is also required to define their contribution to disease progression, alongside conventional treatment assessed according to the international guidelines. Finally, a regulatory intervention to better define the right management of nutraceuticals in cardiovascular disease should increase their impact and credibility in clinical practice. 

## Figures and Tables

**Figure 1 nutrients-13-00257-f001:**
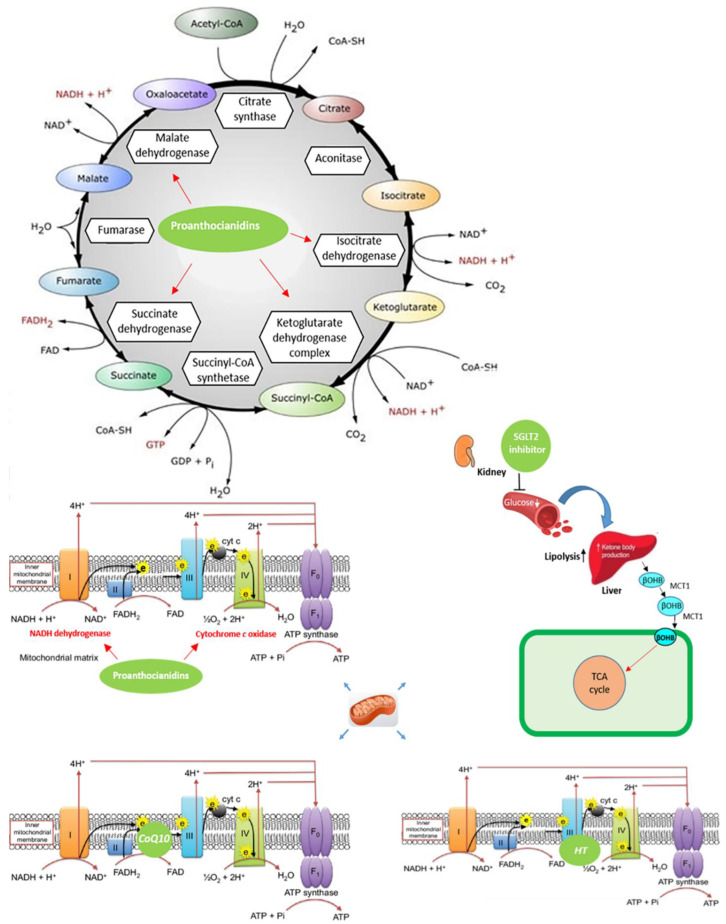
Effects of nutraceuticals on the mitochondrial function in heart diseases. Coenzyme Q10 (CoQ10) plays a critical role in adenosine triphosphate (ATP) generation by accepting electrons from complexes I and II and transporting them to complex III of the mitochondrial electron transport chain. Moreover, CoQ10 is involved in the protons’ transfer in the inner mitochondrial membrane, called the proton motive Q-cycle, leading to the free movement of protons through the internal mitochondrial membrane; sodium–glucose cotransporter (SGLT2) inhibitors increase the amount of ketone bodies; proanthocyanidins significantly increase the activities of mitochondrial enzymes (isocitrate dehydrogenase, succinate dehydrogenase, malate dehydrogenase and α-ketoglutarate dehydrogenase) and respiratory chain enzymes (nicotinamide adenine dinucleotide (NADH) dehydrogenase and cytochrome c oxidase); hydroxytyrosol (HT) is able to improve the integrity of complex-III of the mitochondrial electron transport chain.

**Figure 2 nutrients-13-00257-f002:**
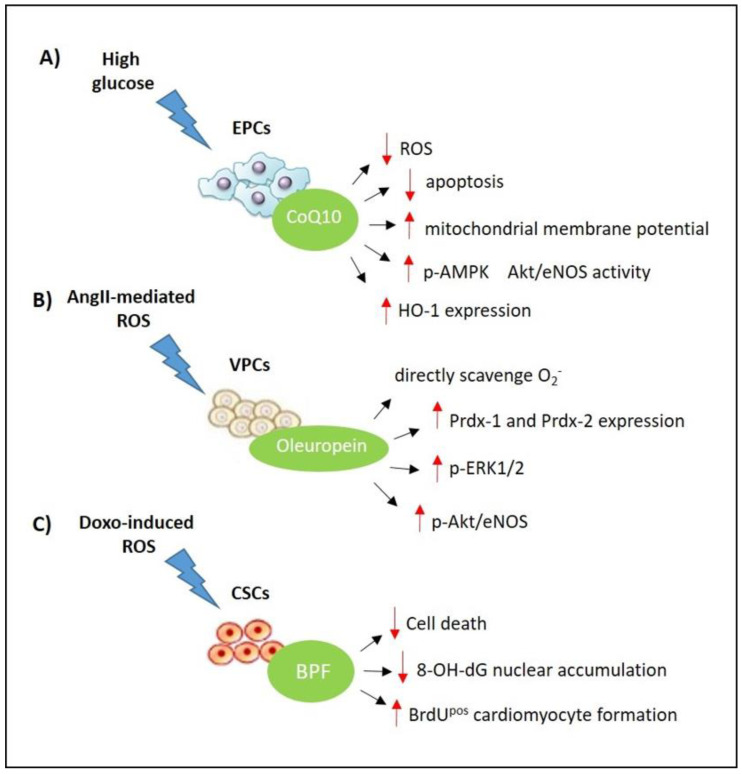
Effects of nutraceuticals on the stem cell compartment in heart diseases. (**A**) CoQ10 inhibits high glucose-induced endothelial progenitor cell (EPC) dysfunction and death via modulation of the 5′ adenosine monophosphate-activated protein kinase (AMPK) pathway, upregulating endothelial nitric oxide synthase (eNOS) activity and heme oxygenase-1 (HO-1) expression; (**B**) oleuropein attenuates AngII-mediated oxidative stress in vascular progenitor cells (VPCs) though direct scavenging activity and regulates the ERK1/2-Prdx and Akt/eNOS signalling pathway; (**C**) the bergamot polyphenolic fraction (BPF) protects endogenous CSCs against DOXO-induced cardiotoxicity though its direct antioxidant properties and stimulates CSC activation and differentiation in newly formed cardiomyocytes.

**Table 1 nutrients-13-00257-t001:** Protective effects of nutritional components in clinical trials.

Bioactive Component	Clinical Trials	Study Duration	Dosage Supplement	Properties	References
Coenzyme Q10	- Meta-analysis of 13 randomised controlled trials - Includes 395 participants (49.8 to 68.0 years) At baseline: - Blood CoQ_10_ concentration 0.61–1.01 μg/mL. - EF 22–46%. - NYHA functional class 2.3–3.4	4–28 weeks	60–300 mg/day	- Pooled mean net increase in blood CoQ_10_ concentration: 1.4 μg/mL (95% CI: 1.1, 1.7 μg/mL) - Pooled mean net increase in EF: 3.67% (95% CI: 1.60%, 5.74%) - Study-specific changes in EF CoQ_10_ supplementation vs. placebo: −3.0% (95% CI: −10.7%, 4.7%) to 17.8% (95% CI: 7.2%, 28.4%) - Pooled mean net change in NYHA classification: −0.30 (95% CI: −0.66, 0.06)	[21]
	- A randomised controlled multicenter trial Q-SYMBIO - Includes 420 patients with moderate to severe HF At baseline: - Chronic HF in NYHA functional class III or IV - Typical symptoms and signs of HF - No specific cut-point related to EF	Primary short-term endpoints (16 weeks): - NYHA functional class and functional status - VAS for symptoms - 6MWT - echocardiography - Serum samples for CoQ10 and NT-proBNP. Primary long-term endpoint (106 weeks): - Morbidity and mortality - VAS for symptoms - MACE	100 mg 3 times/day	After 16weeks: - The level of serum CoQ10-treated group increased to about 3 times the baseline value - NT-proBNP reduction (mean value of 384 pg/mL (20%) vs. 199 pg/mL (12%) After 106 weeks: - Fewer MACE (*N* = 30, 15% vs. *N* = 57, 26%) - Improved NYHA functional classification (*N* = 86, 58% vs. *N* = 68, 45%) - At least 1 grade of improvement in NYHA functional class - Reduced serum NT-proBNP (mean value of 1137 pg/mL; 60% vs. mean value 881 pg/mL; 52%) - Lower total number of cardiovascular deaths (*N* = 18, 9% vs. *N* = 34, 16%), corresponding to a 43% relative reduction. - Lower number of hospital stays for HF (*N* = 17, 8% vs. *N* = 31, 14%) - Lower number of adverse events (mean value 26 (13%) vs. 41 (19%)	[22]
BPF	- A randomised, double-blind, placebo-controlled study - Includes 237 patients: group A (104 patients) with isolated HC (LDL-C levels ≥130 mg/dL) group B (42 patients) with HC/HT group C (59 patients) with mixed HC/HT/HG over 110 mg/dL group D (32 patients) who stopped simvastatin therapy	30 consecutive days	500, 1000 or 1500 mg/day	- In group A, B and C reduction in: - tChol (mean value from 278 mg/mL to 199 mg/mL) - LDL-C (mean value from 188 mg/mL to 126 mg/mL) - TG (mean value from 267 mg/mL to 158 mg/mL) - Increase in HDL-C was observed with the best response in the 10% subjects (−64.6%) - Reduction in blood glucose levels (mean value of −18.9% under 500 mg BPF treatment and −22.4% in 1000 mg BPF treatment) - Group D, 30/32 patients showed a reduction in tChol (mean −25% and LDL-C −27.6%) - Increasing flow-mediated vasodilation	[23]
	- A randomised double-blind, placebo-controlled study - Includes 60 patients suffering from T2DM with: Serum glucose >110 mg/dL)) Mixed hyperlipemia (LDL-C >120 mg/dL and TG >175 mg/dL)	30 consecutive days	650 mg BPF twice/day	- Fasting plasma glucose (from 120 ± 1.6 mg/mL to 98 ± 1.3 mg/mL) - tChol (262 ± 14 mg/dL to 196 ± 12 mg/dL) - LDL-C (175 ± 5.8 mg/mL to 116 ± 3.2 mg/mL) - HDL-C (44 ± 4.1 mg/mL to 48 + 3.8 mg/mL) - Triglycerides (252 ± 9 mg/mL to 170 + 7 mg/mL) - Relevant changes in mean particle diameters for VLDL, LDL, and HDL - Decrease the mean concentration of IDL particles to Increase large LDL and to decrease small LDL. - Increase in total HDL particles	[24]
	- A prospective, open-label, parallel group, placebo- controlled study - Includes 77 patients mixed hypercholesterolemia (LDL-C = 160 mg/dL, TG = 225 mg/dL)	30 consecutive days	- Rosuvastatin (10 and 20 mg/day) - BPF (1000 mg/day) - BPF+rosuvastatin (10 mg/day)	- tChol (mean value from 278 ± 4 mg/dL to 191 ± 5 mg/dL) - LDL-C (mean value from 191 ± 3 mg/dL to 191 ± 3 mg/dL) - TG (mean value from 238 ± 5 mg/dL to 165 ± 3 mg/dL) - HDL-C (mean value from 38 ± 2 mg/dL to 45 ± 4 mg/dL) - Association of BPF+rosuvastatin enhance the hypolipidemic effect of rosuvastatin: - tChol (mean value from 172 ± 3 mg/dL to 195 ± 3 mg/dL) - LDL-C (mean value from 90 ± 4 mg/dL to 115 ± 4 mg/dL), - TG (mean value from 152 ± 5 mg/dL to 200 ± 4 mg/dL) - HDL-C (mean value from 52 ± 4 mg/dL to 42 ± 3 mg/dL) - Reduced urinary MVA excretion - Decreased both MDA levels and LOX-1 expression in PMNs of patients - Enhanced phosphorylation of PKB1 expression in PMNs of patients - The effect of BPF produced a further enhancement of rosuvastatin antioxidant and vasoprotective effects	[25]
Oleuropein, hydroxytyrosol	- An open, controlled, parallel-group, co-twin study - Including 40 borderline hypertensive monozygotic patients (18 and 60 years) with: an untreated SBP >120 mmHg or DBP >80 mmHg at rest	8 weeks	500 mg and 1000 mg/day	- 1000 mg significantly reduces: SBP (mean value 126 ± 9 vs. 137 ± 10) DBP (76 ± 6 vs. 80 ± 10) - Reduction in LDL-C	[26]
	- A double-blind, randomised, parallel and active- controlled clinical study - Includes 232 patients (25 and 60 years) in stage-1 hypertension with SBP of 140–159 mmHg, DBP <90 mmHg or in between 90 and 99 mmHg	- 4-weeks single-blind placebo (is diet-alone) - Run-in period and followed by 8-weeks double-blind treatment period	500 mg twice/day vs. captopril treatment (12.5 mg twice/day)	- significant reduction in: SBP (−11.5 ± 8.6 mmHg vs−13.7 ± 7.6 mmHg in captopril) DBP (−4.8 ± 5.5 mmHg vs. −6.4 ± 5.2 mmHg captopril) - LDL-C (−3.89 ± 19.40 mg/dL vs. 2.14 ± 14.20 mg/dL in captopril) - TG (−11.90 ± 46.17 mg/dL vs. −1.26 ± 43.31 mg/dL in captopril)	[27]
SGLT2-inhibitors	- EMPA-REGOUTCOMEtrial, a cardiovascular outcome trial - Includes 7020 patients (≥18 years) with T2DM at high risk of CV disease with: - BMI ≤ 45 kg/m^2^; - No glucose-lowering therapy in previous 12 weeks -HbA1c 7.0–9.0%, or stable glucose-lowering therapy and HbA1c 7.0–10.0% - Glomerular filtration rate (eGFR) of at least 30 mL/min × 1.73 m^2^ of body-surface area	3 years Primary long-term endpoint: 3P-MACE	10 mg or 25 mg	Primary outcome: - All-cause mortality: HR 0.68 (95% CI 0.57, 0.82; *p* < 0.001) - Incident or worsening nephropathy: HR 0.61 (95% CI 0.53, 0.70; *p* < 0.001) - Lower rates of death from cardiovascular causes (3.7%, vs. 5.9% in te placebo; 38% relative risk reduction; HR 0.62 (95% CI 0.49, 0.77; *p* < 0.001) - Hospitalization for heart failure (2.7% and 4.1%, respectively; 35% relative risk reduction; HR 0.65 (95% CI 0.50, 0.85; *p* = 0.002) - Death from any cause (5.7% and 8.3%, respectively; 32% relative risk reduction)	[28]
	- Declare-TIMI 58 trial - includes 17,160 patients (40 years) with T2DM and risk of atherosclerotic CV disease T2DM; HbA1c ≥6.5%	4, 5 years Primary long-term end-point: - Non-inferiority for 3P-MACE - Composite kidney outcome	10 mg	Co-primary efficacy outcomes−3P-MACE: - lower rate of cardiovascular death or hospitalization for heart failure (4.9% vs. 5.8%; HR, 0.83; 95% CI, 0.73 to 0.95; *p* = 0.005) - Lower rate of hospitalization for heart failure (HR, 0.73; 95% CI, 0.61 to 0.88) kidney composite outcome: - A renal event occurred in 4.3% vs. 5.6% in placebo (HR, 0.76; 95% CI, 0.67 to 0.87) - Death from any cause occurred in 6.2% and 6.6%, respectively (HR, 0.93; 95% CI, 0.82 to 1.04)	[29]

BMI: body mass index; BPF: bergamot polyphenolic fraction; CI: Confidence Interval; EF: ejection fraction; DBP: diastolic blood pressure; HC/HT: hypercholesterolemia and hypertriglyceridemia; HC/HT/HG: hyperlipidemia and glycemia; HDL-C: high density lipoprotein cholesterol; HF: heart failure; HR: hazard ratio; HC: hypercholesterolemia; LDL-C: low density lipoprotein cholesterol; LOX-1: Lectin-like oxidised low-density lipoprotein (LDL) receptor-1; MACE: major adverse cardiovascular events; MDA: malondialdehyde; MVA: mevalonate; NYHA: New York Heart Association; NT-proBNP: N-terminal pro–B-type natriuretic peptide; PMNs: peripheral blood mononuclear cells; PKB: protein kinase B; SBP: systolic blood pressure; tChol: total cholesterol; TG: triglycerides; T2DM: type 2 diabetes mellitus; VAS: visual analogue scale; 6MWT: 6-min walk test; HbA1c: glycated hemoglobin.

**Table 2 nutrients-13-00257-t002:** The main properties of different natural compounds.

Plant	Bioactive Component	Properties	In Vitro/In Vivo Models	Clinical Trials	References
*Brassicaceae family* *Gramineae family*	Coenzyme Q10	Antioxidant and anti-inflammatory activity Key component of METC and in ATP production Bioenergetic effect ↑ p-AMPK ↑ Akt/eNOS activity ↑ HO-1 expression ↑ hemodynamic parameters ↑ LV function ↓ 3-NT ↓ MDA ↓ Nox2 gene expression ↑ Vasodilation ↓ Aldosterone levels ↑ Fatty acid oxidation ↑ VLDL ↓ LDLc/HDLc ↓TC/HDLc ↓ Fibrinogen ↓ SBP ↓ DBP	- EPCs - Isoproterenol- induced HF in rats - Diabetic cardiomyopathy in mice	-HFrEF - Hypertension -T2DM - MetS - Hyperlipideima - MI	[18,19,20,21,22,30,31,32,33,34,35,36,37,38]
*Citrus Bergamia* *Risso et Poiteau*	BPF	↓ Serum glucose, TG, TC, LDL-C, VLDL-C ↑ HDL-C ↑ fecal sterol excretion Re-arrangement of lipoprotein particles ↑ Lipid transfer protein system ↓ pCEH ↑ SOD, catalase ↓ SMC proliferation, LOX-1, p-PKB ↓ ROS, TBARS, MDA, Nitrotyrosine ↑ LV function ↓ pathologic cardiac remodelling ↓ detrimental autophagy ↓ apoptosis ↓ 8OHdG ↑ newly formed myocytes	- eCSCs - Rat neointimal hyperplasia - Hypercholesterolemic diet fed rats - Doxo-induced cardiotoxicity in rats	- Hyperlipemia - MetS - T2DM	[23,24,25,39,40,41,42,43,44,45,46,47]
*Oleaceae family* *(Olea europaea Linn.)*	Oleuropein, hydroxytyrosol	↓ CK-MB, GSSG, TBARS, LDH ↓ MDA, 3-NT, ET-1, IL-1 β, IL-6, TNFα ↑ eNOS ↓ PCs, iNOS ↑ p-Akt, p-AMPK ↑Prdx-1 and Prdx-2 ↓ TC, TG ↑ SOD and GSH activity ↑ integrity of complex III of the METC ↓ infarct size ↓ myocyte apoptosis ↑ LV fuction ↑ pGS3K-β/GS3K-β ↑ Sirt-1, pFOXO3a ↓ myocardial fibrosis ↓ pMEK, pERK1/2, p53, p-IκBα ↑ pSTAT-3 ↓ CYP2E1, OH-1, NF-κB, COX-2 ↓ GRP78, CHOP	- VPCs - CoCl_2_-induced hypoxia in H9C2 cells - ISO-induced MI in rats - Myocardial I/R in rats - Doxo-treated rats - Myocardial I/R in hypercholesterolemic rabbits - Myocardial infarction in rats - T2DM and renal hypertension in rats - Cisplatin-induced kidney injury in mice		[48,49,50,51,52,53,54,55,56,57,58,59,60,61,62,63]
*Malus* *Malus sieversii*	Phlorizin	↑ endothelium-indipendent relaxation ↓ human ox-LDL ↓ postprandial blood glucose rise, HbA1c ↑ urinary glucose excretion ↓ urinary albumin excretion ↓ glycohemoglobin, insulin, TG ↑ muscle GLUT4 ↓ renal GLUT2 ↓ kidney epithelial vacuolization ↓ hepatic glucose production ↑ ketone bodies amount	- Isolated rabbit coronary artery - High-glucose diet in Std ddY mice - Neonatally STZ-induced diabetic rats - STZ-induced diabetic rats		[64,65,66,67,68,69,70,71,72,73]
*Vitis vinifera*	Proanthocyanidins	free radical scavenging activity ↓ TBARS levels ↓ DNA fragmentation ↓ p53 ↑ Bcl-2, Bcl-XL ↓ liver toxicity and DNA fragmentation ↓ hepatocyte apoptosis and necrosis ↓ JNK-1, c-Jun ↓cardiomyocyte apoptosis ↑ post-ischemic cardiac function ↑ mitochondrial IDH, SDH, MDH, α-KGDH ↑ respiratory chain NADH dehydrogenase and CCO	- TPA-induced ROS/RNS in mice - Human oral keratinocytes - AAP-induced liver injury in mice - Myocardial I/R in rats		[74,75,76,77,78,79,80,81,82]
*Zygophyllaceae family* (*Tribulus terrestris* L.)	ferulic acid, diosgenin, saponins	↑Total antioxidant activity ↑ SOD, GPx, CAT activity ↓ MDA ↓ LDH, CK-MB, SGOT, SGPT, Calcium ↑ cell viability ↑ Integrity of mitochondrial PTP ↑ Activity of mitochondrial respiratory complexes ↑ oxygen consumption rate ↑ ATP level ↓ HIF-1α ↑ Mitochondrial OPA1, Mfn1, Mfn2 ↓ Drp1 and Fis1 ↓ Bax, Bad ↑ Bcl-2, p-Akt ↓p-P38, p-JNK ↓ Heart rate ↓ cardiac fibrosis ↓ IL-6, TNFα, IL-1β, MCP-1 ↑ IL-10 ↓ Nuclear traslocation of NF-κB ↑ coronary artery dilation ↑ ECG	- Myocardial ischemia in H9C2 - ISO-induced ischemia in rats	- Angina pectoris	[83]
*Vitis vinifera*	Resveratrol	↑ oxygen consumption rate ↑ SIRT1 and p-AMPK ↓ PCG1α acetylation ↑ PCG1α activity ↑ insulin sensitivity ↑ endothelial function ↑SBP ↑ adiponectin, IL-10 ↓ PAI-1, hsCRP, TNF α, IL-6 ↓ HbA1c, TC, TG ↓ number of angina episodes ↑ LV diastolic function ↓ ANP ↑ LC3II/LC3I ↓p-mTOR, p-p70S6K ↑myocardial ATP content ↓ cleaved-caspase-3	- C2C12 mouse myoblast cells - MEFs (mouse embrionic fibroblasts) - H9C2 cells - Isolated gastrocnemius muscle - HFD-treated mice - KKAy mice - MI in mice and rats - SHRs rats	- Hypertension - CAD - T2DM - Stable angina pectoris - MI	[84,85,86,87,88,89,90,91,92,93]

↑ increase, ↓ decrease. BPF, Bergamot Polyphenolic Fraction; METC, Mitochondrial Electron Transport Chain; ATP, Adenosine Tri-Phosphate; AMPK, 5′ Adenosine Monophosphate-activated Protein Kinase; eNOS, endothelial Nitric Oxide Synthase; H0-1, Heme oxygenase 1; LV, Left Ventricular; 3-NT, Nitrotyrosine; MDA, Malonildialdehyde; Nox2, NADPH oxidase-2; VLDL-C, Very Low Density Lipoprotein Cholesterol; LDL-C, Low Density Lipoprotein Cholesterol; HDL-C, High Density Lipoprotein Cholesterol; TC, Total Cholesterol; SBP, Systolic Blood Pressure; DBP, Diastolic Blood Pressure; TG, Triglycerides; pCEH, pancreatic Cholesterol Ester Hydrolase; SOD, Superoxide Dismutase; SMC, Smooth Muscle Cells; LOX-1, Lectin-type Oxidized LDL receptor 1; PKB, Protein Kinase B; ROS, Reactive Oxigen Species; TBARS, Thiobarbituric Acid Reactive Substances; 8OHdG, 8-Hydroxy-2′-deoxyGuanosine; GPx, Glutathione Peroxidase; CK-MB, Creatine kinase isoenzyme; GSSG, Glutathione disulfide; LDH, Lactate Dehydrogenase; ET-1, Endothelin-1; IL-1β, Interleukin-1β; IL, Interleukin; TNF-α, Tumor Necrosis Factor- α; PCr, Phospho-Creatine; iNOS, inducible Nitric Oxide Synthase; Prdx, Peroxiredoxin; GSH, Glutathione; GS3K-β, Glycogen Synthase Kinase-β; Sirt-1, Sirtuin-1; FOXO3a, Forkhead box O3a; ERK1/2, Extracellular signal-Regulated Kinase 1/2; IκBα, nuclear factor of kappa light polypeptide gene enhancer in B-cells Inhibitor-α; STAT-3, Signal Transducer and Activator of Transcription 3; CYP2E1, Cytochrome P450 Family 2 Subfamily E Member 1; NF-κB, Nuclear Factor kappa-light-chain-enhancer of activated B cells; COX-2, Cyclooxygenase-2; GRP78, Glucose-Regulated Protein 78; CHOP, C/EBP Homologous Protein; ox-LDL, oxidized-Low Density Lipoprotein; HbA1c, glycated Haemoglobin; GLUT, Glucose Transporter; Bcl-2, B-cell lymphoma 2; Bcl-xL, B-cell lymphoma-extra large; JNK-1, c-Jun N-terminal Kinase; IDH, Isocitrate Dehydrogenase; SDH, Succinate Dehydrogenase; MDH, Malate Dehydrogenase; α-KGDH, *α*-Ketoglutarate Dehydrogenase; NADH, Nicotinamide Adenine Dinucleotide; CCO, Cytochrome C Oxidase; GPx, Glutathione Peroxidase, CAT, Catalase; SGOT, Serum Glutamic Oxaloacetic Transaminase; SGPT, Serum Glutamic Pyruvic Transaminase; PTP, Permeability Transition Pore; HIF-1*α*, Hypoxia-Inducible Factor 1-*α*; OPA1, Optic atrophy Protein-1; Mfn, Mitofusin 1; Drp1,Dynamin-related protein 1; Fis1, mitochondrial Fission 1 protein; Bax, Bcl-2-like protein 4; Bad, Bcl-2 associated agonist of cell death; MCP-1, Monocyte Chemoattractant Protein-1; ECG, electrocardiogram; PCG1α, Peroxisome proliferator-activated receptor gamma Coactivator 1 alpha; PAI-1, Plasminogen Activator Inhibitor; Hs-CRP, High sensitivity C-Reactive Protein; ANP, Atrial Natriuretic Peptide; LC3, Microtubule-associated protein 1A/1B-light chain 3; mTOR, mechanistic Target Of Rapamycin; EPCs, Endothelial Progenitor Cells; HF, Heart Failure; eCSCs, endogenous Cardiac Stem Cells; Doxo, Doxorubicin; VPCs, Vascular Progenitor Cells, ISO, Isoproterenol; MI, Myocardial Infarction; I/R, Ischemia/Reperfusion; T2DM, Type 2 Diabete Mellitus; STD, Streptozotocin; TPA, 12-O-tetradecanoylphorbol-13-acetate; RNS, Reactive Nitrogen Species; AAP, acetaminophen; HFD, High Fed Diet; HFrEF, Heart Failure with reduced Ejection Fraction; MetS, Metabolic Syndrome; CHD, Chronic Heart Disease; CAD, Coronary Artery Disease.

## Data Availability

Data available in a publicly accessible repository. **Aknowledgments:** We would like to thank Yavette Schupe for the editorial support in revising the manuscript.

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
