# Peer review of "Pathophysiological Basis for Nutraceutical Supplementation in Heart Failure: A Comprehensive Review"

_nutrients, 2021, doi:10.3390/nu13010257_

Round 1

Reviewer 1 Report

In this comprehensive review, the authors evaluated the pathophysiological mechanisms involved in the early stages of HF and the potential for timely nutraceutical supplementation to support the failing myocardial cells.

The hypothesis is pertinent and relevant, based on the potential of nutraceutical supplementation in counteracting myocardial dysfunction in HF patients. The manuscript reviews all the evidence supporting nutraceutical supplementation for treating patients with impaired cardiac function. This is an original topic as evidence in patients with HF is scarce.

The paper is well-written and easy to read and understand. however, authors should include a final paragraph discussing  future perspectives and clinical implications for nutraceutical supplementation in HF.

Author Response

The paper is well-written and easy to read and understand. however, authors should include a final paragraph discussing  future perspectives and clinical implications for nutraceutical supplementation in HF.

The revised manuscript, according to the referee’s suggestions, now includes a concluding remarks and future perspectives including clinical implications for nutraceutical supplementation in HF.

Reviewer 2 Report

Mocalle et al. have written a comprehensive manuscript on heart failure and the molecular mechanisms of action of nutraceutical/antioxidant supplementation. The review is informative and the mechanisms/effects of nutraceuticals in alleviating cardiac failure are discussed well. I will accept this manuscript for publication but with a lot of changes. Especially much of the text has seems like it has matches from other texts when I checked for plagiarism. I understand that this is a review however, the authors need to write the content in their own words. Rearranging sentences from other manuscripts will not suffice. Please run this manuscript through plagiarism software like turnitin and make appropriate corrections.

Before this manuscript is accepted, I recommend that the authors make the sentences simpler and remove verbose statements as well. In the abstract alone, there are three grammatical errors (including spelling mistakes). The Figures need to be improved significantly. Especially, Figure 1 is not very readable and the fonts are tiny.   The authors should include some more content on clinical trials involving antioxidant nutraceuticals. I am sure that there are more studies which have shown the positive/negative effects of nutraceutical supplementation.   One of the important questions from my perspective is - do antioxidants always help heart failure patients? I have observed that reductive stress also is a key factor and these antioxidant supplements may also cause/aggravate HF. The authors need to also focus on the negative side of antioxidant supplementation. If clinical trials have been cited in the manuscript, for the sake of clarity, the authors could mention the clinical efficacy of those trials. 

Overall, the manuscript has detailed information, but presentation needs to be improved. 

Reviewer 3 Report

The review by Vincenzo  Mollache and coauthors “Pathophysiological basis for nutraceutical supplementation in heart failure- a comprehensive review” is a very extensive overview off the pathophysiological mechanisms of cardiac and vascular   function disturbances and possible  mechanisms of modification based on experimental and animal models. This part is interesting and covers multiple pathways of cardiac dysfunction

The second part deals with possible nutraceutical interventions  in different cardiovascular disorders including heart failure.  The large text addresses only a few nutrition supplements such as  CoQ10, grape seed extract Olea Europea- L related  antioxidants ,SG LT2 inhibitors reach apple extract and bergamot polyphenolic fraction .  There is no explanation given why those nutritients were selected

My main critical remarks

  1. The introduction should be more to the point of nutrition and heart failure and not an epidemiological overview of HF
  2. In the text more data on the results of clinical trials on nutrition components covered in the review (if available ) in cardiovascular trials of HF  should be given and not only an overview of possible cellular mechanisms derived from multiple experimental data and multiple organ effects. IF data from trials are not available for discussed components -it should be stated clearly . Maybe a table presenting the effects of discussed in the review of all four main nutrition components should be given to the reader. It will present a true clinical status of EBM on nutricients
  3. The text should be more condensed and focused on  heart failure mechanisms and not a general beneficial effects on lowering  cholesterol or hypertensive lowering effects.
  4. The presented fragment and citations below deal with VitD and thiamine deficiency which was not covered in any fragment of this review

Quote

“113 In fact, data collected in patients with HF and reduced ejection fraction (HFrEF) suggest that supplementation is accompanied by a non-significant decrease in mortality and cardiovascular hospitalisations[107,108]. Many other studies were too small or underpowered to  adequately appraise clinical outcomes [107, 108]”

  1. The citations are extensive and in part only slightly linked to the main subject of the article.  Please reduce the number of citations.

Reviewer 4 Report

The review by Mollace, et al. entitled “Pathophysiological Basis for Nutraceutical Supplementation in Heart Failure: a Comprehensive Review” an interesting topic that is much needed in the cardiology and nutrition fields. However, I am sorry to write that it soon became apparent to me that there are likely and significant ethical concerns regarding this article. When cross-checking the citations in Sections 2.1-2.3 alone, I found example after example of citations that were inaccurately represented by the authors. These mis-representations of the original papers include 1) stating findings that were not present, 2) mis-interpreting data and/or conclusions, and 3) confusing basic widely-known facts and processes that were being investigated. There are also several cases in which the authors failed to cite, or incorrectly cite, evidence that is being discussed. And, confusingly, the authors incorrectly comment on the findings of their own previously published paper (Gliozzi, et al., line 262).

Major points:

1.Table 1 is difficult to follow as it looks like everything determined for one plant was merged into one row. Might there be a better way to organize the data to prevent words in narrow columns from getting split into different lines? One possibility could be to designate the first line to the plant species, its common name (to make it more relatable to the reader), and its bioactive compound. Subsequent lines could be independently designated for the system studied (in vitro models, in vivo models, or clinical trials). The Properties and References determined for each system could then be attributed accordingly. Further, the use of italics and underlining is inconsistent in the table and the legend.

2.There are many occasions just in sections 2.1, 2.2 and 2.3 in which cited published works are incorrectly attributed, their conclusions are over-stated, or their findings are mis-represented. Such errors diminish the integrity and trustworthiness of the review. Some cases in point include:
3. Lines 153-154. Was there a different citation than that was intended to support the sentence starting with “In fact,”. The work cited by Reference 127 did not 1) study human heart failure (but rather mouse pressure overload), 2) study MCU open probability (but rather MCU knockdown and pharmacologic effects on cardiac parameters), and 3) did not conclude that MCU activity is somehow reduced in heart failure (but likely increased since inhibition of MCU was found to be cardioprotective).
4.The discussion of PERK/eIF2a/ATF4 mechanisms (lines 196-211) contains many errors:
-This pathway is involved in the unfolded protein response; please add “un” to the word folded in line 197.
-Ref. 144 states that it is unclear if PERK is involved in eIF2a phosphorylation, not “is a mediator.”
-Ref. 141 suggests eIF2a inhibits protein translation by a mechanism independent of (not by) its effects on ATF4 expression.
-Is “instantaneous response” referring to the commonly known integrated stress response pathway?
-Ref. 142 investigates ATF6 (not ATF4) that lies in a different pathway.
-Ref. 148 does not mention Grp78, but rather PERK and ATF4.
-No citations are given for the statements beginning “This effect is counteracted…” (line 204).
-Refs. 145, 149 do not support the claim that plant extract ingredients inhibit Grp78 mediated signaling, etc. (line 209).
5.Shouldn’t Ref. 164 in line 244 more correctly be attributed to Ref. 165?
6.Ref. 166 (line 245) investigates hyperglycemia and not NT-MMP-2. It should therefore be placed at the end of the sentence apart from Ref. 165.
7.Line 255, Ref 166 does not mention TSPO and should be taken out.
8.Lines 256-261. The link of TSPO in diabetic cardiomyopathy and in regulating mitochondrial calcium flux cannot be supported by Ref. 169 (or by other nearby references). In fact, Ref. 169 does not mention diabetes, only indirectly discusses Ca2+ with respect to TSPO and states that “the nonspecific effects of many TSPO ligands on calcium handling and other cellular processes such as contractility and excitability have confounded our ability to pinpoint the direct role of TSPO per se in cardiac pathophysiology.”
9.It appears the authors have incorrectly interpreted their own published work in the paragraph beginning on line 262. Further, the statements that follow are unsupported in the absence of additional correct citations.
-First, after reading Gliozzi, et al. (Ref. 167), I could not find where it was “demonstrated that TSPO contributes to determine ER stress and the translocation of the intracellular NT-MMP-2 into mitochondria.” Rather, the data in this paper shows a correlation and not a testing of the relationship, between TSPO function on ER stress and NT-MMP-2 translocation.
-Are there other publications that prove such a causal relationship?
-Second, what evidence is there to support the logic that “This is likely caused by the imbalance of mitochondrial Ca2+ uptake…” (line 263).
-What data exist that has “demonstrated that the driving force of ventricular dysfunction in the heart of diabetic patients depends on NT-MMP-2 intracellular localization…” (line 265)?

Round 2

Reviewer 2 Report

The authors have addressed my concerns

Author Response

ANSWERS TO THE POINTS RAISED BY THE REFEREE N. 4

  1. The grammar is disjointed in many places and it is suggested the authors seek the assistance of an editor.

Editorial support was asked to expert person which is acknowledged at the end of the manuscript and the English appears much improved

2. Reference that are deleted from the original are still listed in the reference section (e.g. Refs 6, 9, 10, 11, 17).

All the references have been included in the right way in the text and now correspond to the reference list, according to the referee’s suggestions

3. Line 76. CHF is not defined

Abbreviation has been changed according to the referee’s suggestions

4. Line 175. What is meant by the reference annotated with undefined MRS (?)

The reference 22 was included to clarify the contribution of Magnetic Resonance Spectroscopy (MRS) in the detection of non-invasive measurements of high-energy phosphate metabolism in the anterior myocardium of heart patients 

5. Lines 199-201. Refs 35-37 do not have data on the expression of Mfn1 or Mfn2, and do not study SR-mitochondrial Ca2+ microdomains.

References 34,35,36 and 37 have been modified and now correct citation appears in the revised manuscript

6. In lines 192-186, the conclusion of altered levels of FADH and NADH2 by changes in the Krebs cycle, and more specifically citrate synthase, cannot support the idea that regeneration of these metabolites are impaired in HF (Line 206-209).

The conclusion has been modified according to the referee’s suggestions

7. Line 405. Ref 90 does not appear to comment on sGC localization and should be corrected.

The due reference has been included and the text modified accordingly

8. Line 407. It is unclear what is referred to in the statement “replicating the oxidatively modified NO-insensitive sGC in knock out mice [90].” Ref 90 discusses eNOS and Gch1/ApoE knockout mice, but not in relationship to sGC.

The sentence was removed and the ref. 90 changed according to the referees suggestions

9. Line 430-434. (Ref 96) It would be more accurate to say that tissue from patients have increased PDE9A and cGMP-esterase activity. This paper did not appear to directly measure cGMP concentration in human hearts. It also does not appear to correlate any kinase activity with passive stiffness.

Appropriate changes have been included in the text according to the referee’s suggestions

10. Lines 444-462. While Ref 99 is appropriate, Ref 98 is not a correct citation here as it is neither a clinical trial or a review or relate to cardio-protection with SGLT2 inhibitors. Also, these paragraphs allude to many findings that are not directly cited (except for Ref 100) that appear to come from the text of Ref 99. I would suggest either highlighting how this section relies heavily on that review article and/or inserting citations from the primary papers that discovered these findings.

Refs 98 and 99 were updated and the corresponding text was modified according to the referee’s suggestions

11. A citation is somehow missing between Refs 100-101. Ref 100 in the text appears to refer to Ref 101 in the References section. This error continues for at least a dozen subsequent citations.

Citations from 100 to 113 were re-arranged and made homogeneous with the text

12. Please check whether Ref 106 in the reference list is the correct citation for 105 in the text.

13. The first author of Ref 108 (cited as 107 line 506) is Uthman, not Baartscheer. Please correct.

Modified

14. Line 517. Is SGLT2 meant instead of CSGLT2?

Corrected

15. Line 531. Ref 113 is not appropriate for citing Empagliflozin here.

Citations from 100 to 113 were re-arranged and made homogeneous with the text

16. Line 553. Is Ref 119 meant instead of Ref 199?

Modified according to the referee’s suggestions

17. Line 556. Ref 120 is not appropriate as a citation for c-Kit.

Modified according to the referee’s suggestions

18. Line 564. Ref 127 is not an appropriate citation for the aging relationship.

Modified according to the referee’s suggestions

19. Is the new paragraph starting on line 636 out of place? It does not directly relate to the discussion of CoQ10. The terms Proanthocyanidins and HT likely appear first here and are confusing to the reader without definitions.

Modified according to the referee’s suggestions

20. The new lines 655-660 are redundant with lines 648-654.

Modified according to the referee’s suggestions

21. Line 662. Please define ISO for the reader.

Isoproterenol was indicated throughout the manuscript

22. Line 703. Ref 158, one of the authors’ previous publications, is inappropriately represented. This study did not measure HMG-CoA levels.

Modified according to the referee’s suggestions

23. Line 709. Ref 155, one of one of the authors’ previous publications, is inappropriately cited. No mention of such an association could be found.

Modified according to the referee’s suggestions

24. Line 775. Ref 138, one of the authors’ previous publications, is wrongly credited for this finding. This is not appropriate and should be attributed to the original manuscript (from P. Anversa’s group?).

Correct references have been included in the revised manuscript

25. Lines 819-823. Sentences are duplicated.

Modified according to the referee’s suggestions

26. Beginning with line 829, hydroxytyrosol was abbreviated as HT and changed to HXT. HT and HXT are used interchangeably thereafter. A consistent use of abbreviation is preferred.

Modified according to the referee’s suggestions

27. Line 931. Ref 205 is not an appropriate citation for a subsequent work by the authors of Refs 202-204 or regarding the analysis of glucose transporters.

28. Line 935. Are references between the text and reference list offset once again? Is Ref 205 intended (published in 1997) rather than Ref 206 (2000). References do not match for several paragraphs thereafter.

The references and the text of this section have been modified according to the referee’s suggestions

29. Section 3.5.1. Most of what is written in this text cannot be correlated with the reference list. Where are the authors finding this information?

This section was substantially re-written  and the citation are now homogeneous with the text according to the referee’s suggestions

30. Line 984. According to the abstract for Ref 216, this paper is not the correct citation for this sentence.

31. Line 987. Please delete old citation [87].

32. Beginning with citation 218 (line 992) references once again do not match reference list.

33. Lines 992-1009. This section discussing Sato et al (incorrectly cited as Ref 219) is overly detailed in comparison with the rest of the review and contains factual errors. The paper did not state the comparisons listed as “approximately 69%” or “approximately 57%” or “approximately 54%.” How did the authors come up with these percentages?

Line 1013. I do not find any of the reviewer authors listed on any of the references near Ref 220 that is cited as “In our study”.

Line 1019. Ref 216 is incorrect.

Modified

Reviewer 4 Report

The authors have addresses some, but not all, of my original concerns. The original Table is reformatted and is now better divided into two tables. Sections 2.1-2.3 now much better reflect the findings from the original literature. However, I am quite surprised and disappointed the authors did not go on to check the accuracy of the rest of the citations in other sections beyond those pointed out in sections 2.1-2.3. It appears the authors have missed my original point that it was very concerning to find so many errors so early into the review and that the rest of the review should therefore be carefully re-examined. Indeed, many more citation errors were readily found throughout the rest of the manuscript. Sadly, this also includes additional inaccurate citations of their own work (Lines 703, 709, and 775; see below). Why was critical assessment of the manuscript not performed prior to re-submission in order to help out the Reviewers in the review process? The new reference list is jumbled and inconsistent with the citations from the text making verification of the references quite difficult. The following is a list of additional errors, and is likely incomplete. May I appeal to the authors once again to accurately represent the literature on which they write?

  1. The grammar is disjointed in many places and it is suggested the authors seek the assistance of an editor.
  2. Reference that are deleted from the original are still listed in the reference section (e.g. Refs 6, 9, 10, 11, 17).
  3. Line 76. CHF is not defined.
  4. Line 175. What is meant by the reference annotated with undefined MRS (?)
  5. Lines 199-201. Refs 35-37 do not have data on the expression of Mfn1 or Mfn2, and do not study SR-mitochondrial Ca2+ microdomains.
  6. In lines 192-186, the conclusion of altered levels of FADH and NADH2 by changes in the Krebs cycle, and more specifically citrate synthase, cannot support the idea that regeneration of these metabolites are impaired in HF (Line 206-209).
  7. Line 405. Ref 90 does not appear to comment on sGC localization and should be corrected.
  8. Line 407. It is unclear what is referred to in the statement “replicating the oxidatively modified NO-insensitive sGC in knock out mice [90].” Ref 90 discusses eNOS and Gch1/ApoE knockout mice, but not in relationship to sGC.
  9. Line 430-434. (Ref 96) It would be more accurate to say that tissue from patients have increased PDE9A and cGMP-esterase activity. This paper did not appear to directly measure cGMP concentration in human hearts. It also does not appear to correlate any kinase activity with passive stiffness.
  10. Lines 444-462. While Ref 99 is appropriate, Ref 98 is not a correct citation here as it is neither a clinical trial or a review or relate to cardio-protection with SGLT2 inhibitors. Also, these paragraphs allude to many findings that are not directly cited (except for Ref 100) that appear to come from the text of Ref 99. I would suggest either highlighting how this section relies heavily on that review article and/or inserting citations from the primary papers that discovered these findings.
  11. A citation is somehow missing between Refs 100-101. Ref 100 in the text appears to refer to Ref 101 in the References section. This error continues for at least a dozen subsequent citations.
  12. Please check whether Ref 106 in the reference list is the correct citation for 105 in the text.
  13. The first author of Ref 108 (cited as 107 line 506) is Uthman, not Baartscheer. Please correct.
  14. Line 517. Is SGLT2 meant instead of CSGLT2?
  15. Line 531. Ref 113 is not appropriate for citing Empagliflozin here.
  16. Line 553. Is Ref 119 meant instead of Ref 199?
  17. Line 556. Ref 120 is not appropriate as a citation for c-Kit.
  18. Line 564. Ref 127 is not an appropriate citation for the aging relationship.
  19. Is the new paragraph starting on line 636 out of place? It does not directly relate to the discussion of CoQ10. The terms Proanthocyanidins and HT likely appear first here and are confusing to the reader without definitions.
  20. The new lines 655-660 are redundant with lines 648-654.
  21. Line 662. Please define ISO for the reader.
  22. Line 703. Ref 158, one of the authors’ previous publications, is inappropriately represented. This study did not measure HMG-CoA levels.
  23. Line 709. Ref 155, one of one of the authors’ previous publications, is inappropriately cited. No mention of such an association could be found.
  24. Line 775. Ref 138, one of the authors’ previous publications, is wrongly credited for this finding. This is not appropriate and should be attributed to the original manuscript (from P. Anversa’s group?).
  25. Lines 819-823. Sentences are duplicated.
  26. Beginning with line 829, hydroxytyrosol was abbreviated as HT and changed to HXT. HT and HXT are used interchangeably thereafter. A consistent use of abbreviation is preferred.
  27. Line 931. Ref 205 is not an appropriate citation for a subsequent work by the authors of Refs 202-204 or regarding the analysis of glucose transporters.
  28. Line 935. Are references between the text and reference list offset once again? Is Ref 205 intended (published in 1997) rather than Ref 206 (2000). References do not match for several paragraphs thereafter.
  29. Section 3.5.1. Most of what is written in this text cannot be correlated with the reference list. Where are the authors finding this information?
  30. Line 984. According to the abstract for Ref 216, this paper is not the correct citation for this sentence.
  31. Line 987. Please delete old citation [87].
  32. Beginning with citation 218 (line 992) references once again do not match reference list.
  33. Lines 992-1009. This section discussing Sato et al (incorrectly cited as Ref 219) is overly detailed in comparison with the rest of the review and contains factual errors. The paper did not state the comparisons listed as “approximately 69%” or “approximately 57%” or “approximately 54%.” How did the authors come up with these percentages?
  34. Line 1013. I do not find any of the reviewer authors listed on any of the references near Ref 220 that is cited as “In our study”.
  35. Line 1019. Ref 216 is incorrect.

Author Response

ANSWERS TO THE POINTS RAISED BY THE REFEREE N. 4

  1. The grammar is disjointed in many places and it is suggested the authors seek the assistance of an editor.

Reply: Editorial support was asked to expert person which is acknowledged at the end of the manuscript and the English appears much improved

  1. Reference that are deleted from the original are still listed in the reference section (e.g. Refs 6, 9, 10, 11, 17).

Reply: All the references have been included in the right way in the text and now correspond to the reference list, according to the referee’s suggestions

  1. Line 76. CHF is not defined

Reply:Abbreviation has been changed according to the referee’s suggestions

  1. Line 175. What is meant by the reference annotated with undefined MRS (?)

Reply:The reference 22 was included to clarify the contribution of Magnetic Resonance Spectroscopy (MRS) in the detection of non-invasive measurements of high-energy phosphate metabolism in the anterior myocardium of heart patients 

  1. Lines 199-201. Refs 35-37 do not have data on the expression of Mfn1 or Mfn2, and do not study SR-mitochondrial Ca2+ microdomains.

Reply: References 34,35,36 and 37 have been modified and now correct citation appears in the revised manuscript

  1. In lines 192-186, the conclusion of altered levels of FADH and NADH2 by changes in the Krebs cycle, and more specifically citrate synthase, cannot support the idea that regeneration of these metabolites are impaired in HF (Line 206-209).

Reply: The conclusion has been modified according to the referee’s suggestions

  1. Line 405. Ref 90 does not appear to comment on sGC localization and should be corrected.

Reply: The due reference has been included and the text modified accordingly

  1. Line 407. It is unclear what is referred to in the statement “replicating the oxidatively modified NO-insensitive sGC in knock out mice [90].” Ref 90 discusses eNOS and Gch1/ApoE knockout mice, but not in relationship to sGC.

Reply: The sentence was removed and the ref. 90 changed according to the referees suggestions

  1. Line 430-434. (Ref 96) It would be more accurate to say that tissue from patients have increased PDE9A and cGMP-esterase activity. This paper did not appear to directly measure cGMP concentration in human hearts. It also does not appear to correlate any kinase activity with passive stiffness.

Reply: Appropriate changes have been included in the text according to the referee’s suggestions

  1. Lines 444-462. While Ref 99 is appropriate, Ref 98 is not a correct citation here as it is neither a clinical trial or a review or relate to cardio-protection with SGLT2 inhibitors. Also, these paragraphs allude to many findings that are not directly cited (except for Ref 100) that appear to come from the text of Ref 99. I would suggest either highlighting how this section relies heavily on that review article and/or inserting citations from the primary papers that discovered these findings.

Reply: Refs 98 and 99 were updated and the corresponding text was modified according to the referee’s suggestions

  1. A citation is somehow missing between Refs 100-101. Ref 100 in the text appears to refer to Ref 101 in the References section. This error continues for at least a dozen subsequent citations.

Reply: Citations from 100 to 113 were re-arranged and made homogeneous with the text

  1. Please check whether Ref 106 in the reference list is the correct citation for 105 in the text.
  2. The first author of Ref 108 (cited as 107 line 506) is Uthman, not Baartscheer. Please correct.

Reply: Modified

  1. Line 517. Is SGLT2 meant instead of CSGLT2?

Reply: Corrected

  1. Line 531. Ref 113 is not appropriate for citing Empagliflozin here.

Reply: Citations from 100 to 113 were re-arranged and made homogeneous with the text

  1. Line 553. Is Ref 119 meant instead of Ref 199?

Reply: Modified according to the referee’s suggestions

  1. Line 556. Ref 120 is not appropriate as a citation for c-Kit.

Reply: Modified according to the referee’s suggestions

  1. Line 564. Ref 127 is not an appropriate citation for the aging relationship.

Reply: Modified according to the referee’s suggestions

  1. Is the new paragraph starting on line 636 out of place? It does not directly relate to the discussion of CoQ10. The terms Proanthocyanidins and HT likely appear first here and are confusing to the reader without definitions.

Reply: Modified according to the referee’s suggestions

  1. The new lines 655-660 are redundant with lines 648-654.

Reply: Modified according to the referee’s suggestions

  1. Line 662. Please define ISO for the reader.

Reply: Isoproterenol was indicated throughout the manuscript

  1. Line 703. Ref 158, one of the authors’ previous publications, is inappropriately represented. This study did not measure HMG-CoA levels.

Reply: Modified according to the referee’s suggestions

  1. Line 709. Ref 155, one of one of the authors’ previous publications, is inappropriately cited. No mention of such an association could be found.

Reply: Modified according to the referee’s suggestions

  1. Line 775. Ref 138, one of the authors’ previous publications, is wrongly credited for this finding. This is not appropriate and should be attributed to the original manuscript (from P. Anversa’s group?).

Reply: Correct references have been included in the revised manuscript

  1. Lines 819-823. Sentences are duplicated.

Reply: Modified according to the referee’s suggestions

  1. Beginning with line 829, hydroxytyrosol was abbreviated as HT and changed to HXT. HT and HXT are used interchangeably thereafter. A consistent use of abbreviation is preferred.

Reply: Modified according to the referee’s suggestions

  1. Line 931. Ref 205 is not an appropriate citation for a subsequent work by the authors of Refs 202-204 or regarding the analysis of glucose transporters.
  2. Line 935. Are references between the text and reference list offset once again? Is Ref 205 intended (published in 1997) rather than Ref 206 (2000). References do not match for several paragraphs thereafter.

Reply: The references and the text of this section have been modified according to the referee’s suggestions

  1. Section 3.5.1. Most of what is written in this text cannot be correlated with the reference list. Where are the authors finding this information?

Reply: This section was substantially re-written  and the citation are now homogeneous with the text according to the referee’s suggestions

  1. Line 984. According to the abstract for Ref 216, this paper is not the correct citation for this sentence.
  2. Line 987. Please delete old citation [87].
  3. Beginning with citation 218 (line 992) references once again do not match reference list.
  4. Lines 992-1009. This section discussing Sato et al (incorrectly cited as Ref 219) is overly detailed in comparison with the rest of the review and contains factual errors. The paper did not state the comparisons listed as “approximately 69%” or “approximately 57%” or “approximately 54%.” How did the authors come up with these percentages?
  5. Line 1013. I do not find any of the reviewer authors listed on any of the references near Ref 220 that is cited as “In our study”.
  6. Line 1019. Ref 216 is incorrect.

Reply: All the points have been modified according to the referee's suggestions
